# ARROW: An Adaptive Rollout and Routing Method for Global Weather Forecasting

**Jindong Tian[1], Yifei Ding[1], Ronghui Xu[1], Hao Miao[2], Chenjuan Guo[1], Bin Yang[1]***
[1]East China Normal University, [2]The Hong Kong Polytechnic University
{jdtian,yfding,rhxu}@stu.ecnu.edu.cn, {hao.miao}@polyu.edu.hk
{cjguo,byang}@dase.ecnu.edu.cn

## Abstract

Weather forecasting is a fundamental task in spatiotemporal data analysis, with broad applications across a wide range of domains. Existing data-driven forecasting methods typically model atmospheric dynamics over a fixed short time interval (e.g., 6 hours) and rely on naive autoregression-based rollout for long-term forecasting (e.g., 138 hours). However, this paradigm suffers from two key limitations: (1) it often inadequately models the spatial and multi-scale temporal dependencies inherent in global weather systems, and (2) the rollout strategy struggles to balance error accumulation with the capture of fine-grained atmospheric variations. In this study, we propose ARROW, an Adaptive-Rollout Multi-scale temporal Routing method for Global Weather Forecasting. To contend with the first limitation, we construct a multi-interval forecasting model that forecasts weather across different time intervals. Within the model, the Shared-Private Mixture-of-Experts captures both shared patterns and specific characteristics of atmospheric dynamics across different time scales, while Ring Positional Encoding accurately encodes the circular latitude structure of the Earth when representing spatial information. For the second limitation, we develop an adaptive rollout scheduler based on reinforcement learning, which selects the most suitable time interval to forecast according to the current weather state. Experimental results demonstrate that ARROW achieves state-of-the-art performance in global weather forecasting, establishing a promising paradigm in this field. The code is available at: https://github.com/decisionintelligence/ARROW.

## 1 Introduction

Global Weather forecasting (GWF) is a fundamental task in various domains (Guo et al., 2014; Hettige et al., 2024; Tian et al., 2025; Lai et al., 2026; 2025). It plays a vital role in supporting decision-making (Yuan et al., 2026) for individual travel planning as well as for key sectors such as energy (Hu et al., 2024; 2026), economics (Mei et al., 2025), and agriculture (Qiu et al., 2025).

Traditional weather forecasting primarily relies on numerical weather prediction methods that solve Partial Differential Equations in atmospheric dynamics. However, these models are computationally expensive when applied to GWF. Consequently, data-driven models (Liu et al., 2024a;b; Zhang et al., 2026) have emerged as an alternative, leveraging historical data to approximate atmospheric dynamics (Rasp & Thuerey, 2021a; Rasp et al., 2020; Ling et al., 2024). A widely adopted strategy in data-driven weather forecasting methods is the *iterative prediction paradigm*. In this approach, the lead time is divided into shorter intervals, and a specific single interval forecasting model (SIFM) is pre-trained for each interval. Predictions are subsequently generated in an autoregressive manner, and cumulative errors are mitigated through fine-tuning. For example, as shown in Fig. 1(a), to predict the weather with a lead time 138h, it repeats a SIFM (with interval 6h) 23 times. This strategy reduces training difficulty and better aligns with the incremental evolution of atmospheric processes, thereby alleviating the challenge of directly predicting atmospheric dynamics over long horizons. Despite the significant advancements and promising results achieved by data-driven methods, two fundamental limitations remain:

---

*Corresponding author

**Insufficient modeling of spatiotemporal dependencies.** The dynamics in weather system across different time intervals are inherently correlated, governed by the same underlying physical principles and exhibiting multi-scale interactions (Frank et al., 2024). However, existing methods (Bi et al., 2023; Du et al., 2025) often train an independent SIFM for each time interval. This isolation not only ignores the intrinsic relationships between forecasts at different temporal scales, but also increases computational overhead and training complexity, as each SIFM must be optimized and maintained independently. Beyond temporal dependencies, spatial dependencies are equally critical for accurate global weather representation. However, existing Transformer-based models often treat the Earth as an "flat image", failing to incorporate its spherical geometry. This simplification introduces distortions in representing global atmospheric phenomena. Effectively unifying the modeling of these spatiotemporal dependencies remains a central challenge.

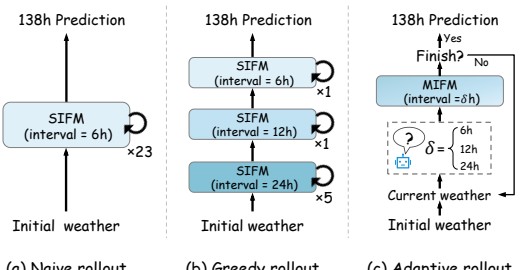

Figure 1: Different rollout scheme in weather forecasting. (a) SIFM with naive rollout. (b) Three SIFMs with greedy rollout. (c) Multi-interval forecasting model (MIFM) with adaptive rollout.

**Inflexible autoregression rollout scheme.** Atmospheric evolution is inherently non-uniform, characterized by intermittent abrupt transitions and periods of gradual change. An effective autoregressive strategy should therefore balance two objectives: leveraging larger time intervals to mitigate error accumulation during stable phases, and employing finer intervals to capture rapid transitions accurately. However, current methods predominantly rely on inflexible autoregression rollout schemes. For example, FourcastNet (Pathak et al., 2022) employs only a SIFM with 6-hour interval applied iteratively (Fig. 1(a)). Pangu-weather (Bi et al., 2023) uses multiple SIFMs of different intervals (e.g., 6-hour, 12-hour, 24-hour) in a predefined greedy rollout, selecting the SIFM with largest available interval at each step (Fig. 1(b)). While capable of generating forecasts autoregressively, these inflexible schemes fail to adapt to the non-uniform nature of atmospheric evolution. Thus, there is a need for an adaptive rollout strategy (Fig. 1(c)) that dynamically selects time intervals based on real-time atmospheric states, enabling improved responsiveness to both short-term changes and long-term trends.

In this paper, we propose **ARROW**, an **A**daptive **R**ollout and **RO**uting method for global **W**eather forecasting. To address the first limitation, we propose a novel multi-interval forecasting model (MIFM) that enables capturing the multi-scale temporal correlations and circular spatial dependencies across different time intervals within a unified model. It has two key components: Shared-Private Mixture-of-Experts (S&P MoE) and Ring Positional Encoding (RPE). Given that the same meteorological phenomena may evolve across different time scales, the S&P MoE leverages shared experts to model variations common to all intervals, while private experts capture interval-specific patterns of atmospheric dynamics. Through its routing mechanism, knowledge of temporal variations at different scales can be effectively exchanged among experts. By introducing two auxiliary losses, the S&P MoE not only promotes specialization of individual experts but also preserves a balanced contribution across all experts. Spatially, RPE encodes the Earth's circular latitude structure within ARROW's tokenizer, enabling a more faithful representation of global atmospheric phenomena.

To overcome the second limitation, we design an Adaptive Rollout Scheduler (AR Scheduler) that selects appropriate rollout strategies based on the initial conditions (e.g., current weather states). This mechanism, similar to adaptive solvers of PDEs in atmospheric dynamics (Hutchinson, 2007), enhances the model's ability to adapt to rapidly evolving atmospheric processes. Within the AR Scheduler, we construct a weather forecasting environment with corresponding state, action, and reward spaces. Q-learning is employed to learn the optimal adaptive rollout strategy for the pretrained MIFM. However, multi-step fine-tuning guided by this strategy alters the MIFM parameters, rendering the original strategy suboptimal. Therefore, we integrate the Q-learning with multi-step fine-tuning in an alternating optimization paradigm. This integration allows ARROW to effectively suppress cumulative errors while learning the corresponding optimal policy. In summary, our contributions can be summarized as follows:

- We propose ARROW, an Adaptive-Rollout Routing method for global weather forecasting. ARROW formulates the adaptive rollout strategy as a decision-making problem that balances error accumulation with the accurate capture of fine-grained atmospheric dynamics.
- We introduce an MIFM that enables forecasts across different time intervals. Within MIFM, the S&P MoE captures evolving weather variations across multiple time scales, while the RPE encodes spatial dependencies arising from the Earth's spherical geometry.
- Experimental results demonstrate that ARROW achieves state-of-the-art performance in global weather forecasting, with overall improvements of approximately **10%** in both RMSE and ACC.

## 2 PRELIMINARIES

### 2.1 PROBLEM STATEMENT

Global weather forecasting aims to predict the future weather state, given the initial weather state $X_0 \in \mathbb{R}^{V \times H \times W}$. Here, $V$ denotes the number of meteorological variables; $H$ and $W$ represent the spatial resolution of the Earth under the equirectangular projection, determined by the granularity of the global grid. We introduce a MIFM, denoted as $f(\cdot; \delta)$, training on a set of short intervals to predict weather changes $\Delta_\delta = X_\delta - X_0$ rather than the absolute future state $X_\delta$. To obtain forecasts at the target lead time, we apply an autoregression rollout along a specified trajectory. For example, given the rollout trajectory $\Gamma = (6h, 18h, .., 114h, 138h)$ for the lead time 138h, the forecast is obtained as: $\hat{X}_{138h} = f(\hat{X}_{114h}; 24h) + \hat{X}_{114h}, ..., \hat{X}_{18h} = f(\hat{X}_{6h}; 12h) + \hat{X}_{6h}, \hat{X}_{6h} = f(X_0; 6h) + X_0$.

### 2.2 RELATED WORK

**Numerical Weather Prediction.** Numerical Weather Prediction (NWP), the traditional method for weather forecasting since the 1950s, is based on curated partial differential equations (PDEs) derived from atmospheric dynamics (Lynch, 2008; Hurrell et al., 2013). Adaptive time-stepping methods dynamically adjust the temporal resolution according to the Courant–Friedrichs–Lewy condition, improving computational efficiency while ensuring numerical stability (Hutchinson, 2007). Mesh-adaptive techniques refine the spatial mesh based on local meteorological conditions, enhancing the model's ability to capture fine-scale atmospheric variations (Skamarock et al., 1989). Although NWP methods are grounded in the first principles of atmospheric physics, they still face significant challenges like systematic errors and computational complexity (Brotzge et al., 2023; Magnusson & Källén, 2013). Most critically, more observation data cannot be directly leveraged to improve forecast accuracy, as NWP fundamentally depends on refined PDEs, accurate parameterizations, and numerical solvers.

**Data-driven Weather Forecasting.** Recent advances in deep learning (Xu et al., 2023; Ouyang et al., 2024; Zhou et al., 2026) have prompted a shift toward data-driven models for global weather forecasting (Rasp & Thuerey, 2021b; Weyn et al., 2020). These methods adopt an *iterative prediction paradigm* (as described in Section 1), which typically consists of two stages: one-step pre-training and multi-step fine-tuning. The core of this paradigm lies the construction of One-Step Forecasting Model (OSFM). Graph-based methods (Keisler, 2022; Lam et al., 2023; Oskarsson et al., 2024) capture spatiotemporal dependencies in the atmosphere through message passing. Transformer-based approaches (Nguyen et al., 2023; Bi et al., 2023) treat the Earth as a 2D or 3D sequence and model atmospheric variations via attention mechanisms. Subsequent innovations, such as Neural ODE (Verma et al., 2024) and Neural Operators (Pathak et al., 2022), have further accelerated progress in this field. To mitigate error accumulation during autoregression (as shown in Fig. 1), most methods apply multi-step fine-tuning to the OSFM. Fengwu (Chen et al., 2023) and Aurora (Bodnar et al., 2025) introduce replay buffer to adapt to input errors during the autoregression. However, during inference, existing approaches still rely on an inflexible rollout to obtain predictions at the target lead time regardless of different initial weather state. To the best of our knowledge, no existing data-driven method consider the adaptive rollout in the weather forecasting.

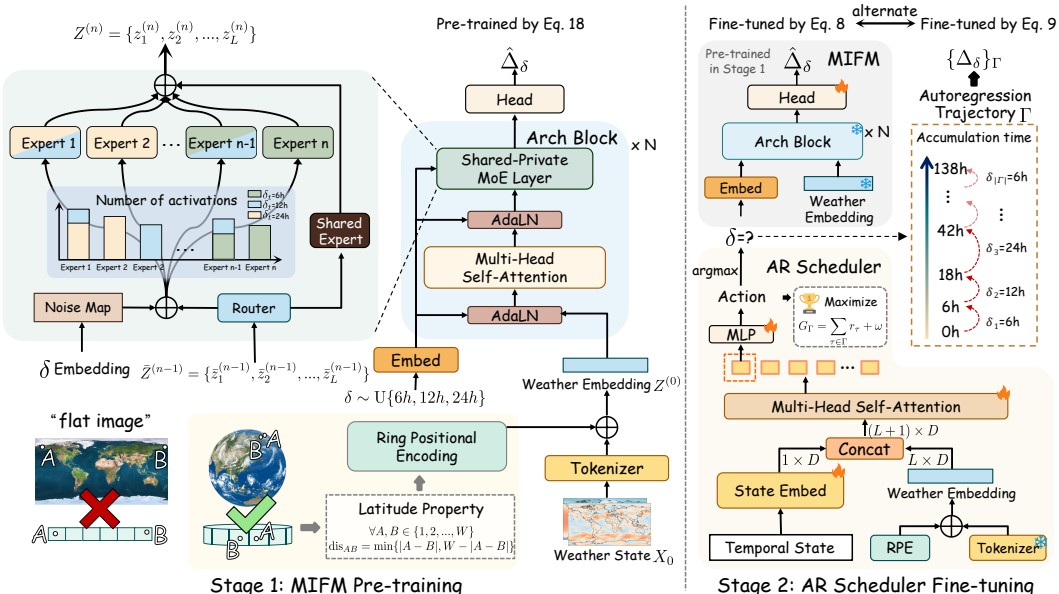

Figure 2: The overall framework of ARROW consists of two stages. One-step pre-training of the Multi-Interval Forecasting Model captures atmospheric dynamics at different intervals. Then, multi-step fine-tuning of the Adaptive Rollout Scheduler enables adaptive rollout under different lead times.

# 3 METHODOLOGY

## 3.1 MODEL OVERVIEW

We propose ARROW, an adaptive rollout and routing method for global weather forecasting. As shown in Fig. 2, the training process comprises two stages: one-step pre-training for the **Multi-Interval Forecasting Model**, followed by multi-step fine-tuning for **Adaptive Rollout Scheduler**.

- **Multi-Interval Forecasting Model (Pre-training).** To capture both spatial and multi-scale temporal dependencies, we introduce and pre-train an MIFM (as shown on the left side of Fig. 2) to generate accurate short-term one-step forecasts across different time intervals. Specifically, the weather state $X_0$ is first processed by a weather-specific Tokenizer. Next, we introduce a Ring Positional Encoding (RPE), which captures the circular structure of the Earth, to encode the spatial information of weather states and generate the corresponding weather embedding. The resulting weather embedding sequence $Z^{(0)}$, together with the embedded time interval $\delta$, is passed into $N$ Arch blocks to produce $Z^{(N)}$. Each Arch block is equipped with a Shared-Private Mixture-of-Experts, enabling the model to capture both shared patterns and scale-specific characteristics across different time intervals. Finally, a head module reconstructs the output $Z^{(N)}$ into a spatial format, yielding the predicted weather state $X_\delta \in \mathbb{R}^{V \times H \times W}$ for the specified time interval.

- **Adaptive Rollout Scheduler (Fine-tuning).** To balance error accumulation and fine-grained atmospheric evolution, we construct an Adaptive Rollout Scheduler (AR Scheduler, as shown on the right side of Fig. 2). The scheduler separately encodes temporal state and current weather state, and then fuses their embeddings. By maximizing the return of AR Scheduler, it determines the appropriate time interval to process. This interval is then passed into the MIFM trained in Stage 1 to generate a one-step forecast. By iteratively invoking the scheduler, the lead-time prediction $X_T$ can be obtained along the autoregression trajectory. Through alternating multi-step fine-tuning of the MIFM and policy learning for the AR Scheduler, ARROW effectively suppresses cumulative errors under the learned policy.

## 3.2 MULTI-INTERVAL FORECASTING MODEL

### 3.2.1 RING POSITIONAL ENCODING

After tokenizing the initial weather state $X_0$ using weather-specific tokenization (Nguyen et al., 2023; 2024), we obtain a sequence $\tilde{Z} \in \mathbb{R}^{L \times D}$, where $L = h \times w$ denotes the number of tokens ($h = H/P$, $w = W/P$ are the token resolutions under patch size $P$) and $D$ is the embedding dimension. We propose Ring Positional Encoding (RPE), which enables ARROW to account for the Earth's latitude property (as shown on the left side of Fig. 2). For any location $k : (p_k^x, p_k^y)$ on an $h \times w$ 2D grid sequence, its positional encoding is given by:

$$\text{RPE}_{k,4i} = sin(\frac{p_k^x}{w} \cdot 2\pi i) \cdot \frac{w}{4i}, \quad \text{RPE}_{k,4i+1} = cos(\frac{p_k^x}{w} \cdot 2\pi i) \cdot \frac{w}{4i}, \tag{1}$$

$$\text{RPE}_{k,4i+2} = sin(\frac{p_k^y}{w} \cdot 2\pi i) \cdot \frac{w}{4i}, \quad \text{RPE}_{k,4i+3} = cos(\frac{p_k^y}{w} \cdot 2\pi i) \cdot \frac{w}{4i}, \tag{2}$$

where $i = 1, ..., \frac{D}{4}$. The details of the 1D RPE and the corresponding property proofs are provided in Appendix B. Note that the properties of Eq. 12 do not apply to the $4i$ and $4i + 1$ dimensions, as longitude does not exhibit circularity after equirectangular projection. Then, weather embedding is obtained as: $Z^{(0)} = \tilde{Z} + \text{RPE}$.

### 3.2.2 SHARED-PRIVATED MIXTURE OF EXPERTS

The design of the Adaptive Layer Normalization (AdaLN) block enables the Transformer to incorporate the time interval $\delta$ as a conditional input. To further capture both shared and specific characteristics of atmospheric dynamics across different intervals, we propose a Shared-Private Mixture-of-Experts that captures (i) global commonalities across all time intervals, (ii) the unique features specific to each time interval, and (iii) the partial commonalities shared among subsets of time intervals.

In the $n$-th Arch Block's S&P MoE, each token $\bar{z}_l^{(n-1)}$ at position $l$ in the representation sequence $\bar{Z}^{(n-1)}$ is routed to $M$ private feed-forward networks (FFNs) based on a gating value $g'$, as defined by the following equation:

$$z_l^{(n)} = \sum_{m=1}^{M} g'_{m,l} \cdot E_m^p(\bar{z}_l^{(n-1)}) + E^s(\bar{z}_l^{(n-1)}), \tag{3}$$

where $E^p$ is the private FFN, $E^s$ is the shared FFN. The gate value $g'_l$ for token $z_l$ is obtained by perturbing the original gate score $s_l$ with noise $b_l^\delta$, as follows:

$$s_l = \text{Sigmoid}(\bar{z}_l \cdot W_{gate}), \qquad b_l^\delta = \text{Sigmoid}(\bar{z}_l \cdot W_{noise,\delta}), \tag{4}$$

$$g_{m,l} = \begin{cases} s_{m,l} & s_{m,l} + b_{m,l}^\delta \in \text{Top-k}(s_l + b_l^\delta) \\ 0 & \text{otherwise} \end{cases}, \qquad g'_l = \text{Softmax}(g_l). \tag{5}$$

In this formulation, $s_l$ and $b_l^\delta$ represent discrete distributions. By adjusting $b_l^\delta$, the routing behavior of $z_l$ can be effectively controlled. It is desirable for the private experts to capture interval–specific features while maintaining load balance. To achieve this, we introduce two auxiliary cross-entropy loss functions, as defined below:

$$P_{\delta_j} = \text{softmax}(\sum_l b_l^{\delta_j}), \quad \text{H}(P, Q) = -\sum p_i \cdot log(q_i), \tag{6}$$

$$\text{aux-loss}_1 = \sum_{i<j} \text{H}\left(P_{\delta_i}, P_{\delta_j}\right), \quad \text{aux-loss}_2 = \text{H}\left(\text{softmax}(\sum P_\delta), \mathcal{U}(E_1, ..., E_M)\right). \tag{7}$$

Here, $H(\cdot, \cdot)$ denotes the cross-entropy loss, which measures the similarity between two probability distributions. The two auxiliary losses in Eq. 7 serve the following purposes:

**Individual characteristics.** The aux-loss$_1$ in Eq. 7 computes the divergence between the noise distributions of different time intervals. A greater divergence is encouraged to promote interval-specific learning and ensure that each private expert captures unique characteristics.

**Global Balance.** The aux-loss$_2$ in Eq. 7 computes the distribution formed by aggregating all noise distributions. To maximize the utilization of all private FFNs, we encourage this aggregate distribution to approximate a uniform distribution. This promotes balanced routing, a common challenge in MoE.

By jointly optimizing with a randomized dynamics forecasting loss $\mathcal{L}_\delta$ (Eq. 17, see Appendix C for details), MIFM can be pre-trained to handle all time intervals within a unified model.

## 3.3 Adaptive Rollout Scheduler

Although an MIFM can effectively capture the atmospheric dynamics associated with different time intervals, weather forecasting also requires a robust autoregressive rollout strategy to balance error accumulation and fine-grained atmospheric evolution.

In this paper, we propose an Adaptive Rollout Scheduler $\mathfrak{S}$ (AR Scheduler) and a weather forecasting environment $\Psi$, which includes the pretrained ARROW model and the same weather dataset used in Stage 1. By interacting with the environment, the AR Scheduler learns to generate an autoregressive trajectory $\Gamma = \{\tau_1, \tau_2, \ldots, \tau_t, \ldots, \tau_{|\Gamma|}\}$ towards the target lead time (as illustrated in Fig. 2, *Autoregression Trajectory*). At each step $t$, the AR Scheduler adaptively selects a time interval $\delta_t$ based on the predicted current weather state $\hat{X}\tau_{t-1}$, its corresponding temporal information. The prediction error for $\hat{X}_{\tau_t}$ can then be computed by comparing it to $X_{\tau_t}$ retrieved from the weather dataset. Based on this setup, the environment $\Psi$ is formally defined as follows.

**State Space ($\mathcal{S}$).** For any step $t$ in the autoregression, the state space for adaptive rollout forecasting consists of the weather state $\hat{X}_{\tau_{t-1}}$, the date&time (e.g., August 12, 2018, 6:00 AM), and the temporal position $\tau_t$ in trajectory, which includes travel time, remaining time, and target lead time. Among these, the weather embedding $Z_{\tau_{t-1}}^{\text{weather}} \in \mathbb{R}^{L \times D}$ is $Z^{(0)}$ in the pre-trained ARROW, while the temporal embedding $Z_{\tau_{t-1}}^{\text{time}} \in \mathbb{R}^{1 \times D}$ is generated by the temporal embedding module. (Illustrated in the right side of Fig. 2)

**Action Space ($\mathcal{A}$).** The action space is a discrete set consisting of the available time intervals used in the MIFM. As illustrated in Fig. 1, the action space in ARROW is $\{6h, 12h, 24h\}$.

**Reward Function ($\mathcal{R}$).** The reward $r_\tau \sim r(s_\tau, a_\tau)$ is defined as the negative latitude-weighted root mean square error of the prediction at each step in the trajectory. To prevent excessive cumulative error caused by overly long rollout trajectory, a penalty term $\omega$ is added to the reward at each step, resulting in $r_\tau + \omega$. Accordingly, the return of an arbitrary trajectory $\Gamma$ is defined as: $G_\Gamma = \sum_{\tau \in \Gamma} r_\tau + \omega$.

To reduce the complexity of trajectory combinations and enhance generalization through neural networks, we adopt a Deep Q-Network (DQN) to estimate the value function over the state–action space in weather forecasting. As illustrated in Fig. 2, the temporal embedding $Z_\tau^{\text{time}}$ and the weather embedding $Z_\tau^{\text{weather}}$ are concatenated and passed through a Multi-Head Self-Attention module. The last token is then fed into the head to produce a value estimate for each (state, action) pair. The DQN is optimized using Eq. 8, and the resulting policy corresponds to the environment $\Psi$ equipped with the pre-trained ARROW, denoted as $\pi_{\text{ARROW-P}}$.

$$\mathcal{L}_\Psi = \mathbb{E}_{(S,A,R,S') \sim D_\Psi} \left[ \left( R + \gamma \cdot \max_{a \in \mathcal{A}(S')} q_{\text{target}}(S', a) - q_{\text{main}}(S, A) \right)^2 \right], \tag{8}$$

where $S'$ denotes the next state, and $(S, A, R, S')$ samples from the distribution related to the environment $\Psi$. The function $q(s, a)$ is to estimate the state–action value. Specifically, $q_{\text{target}}$ refers to the target network used to compute the temporal difference target, with frozen parameters, while $q_{\text{main}}$ denotes the active main network, which is updated regularly and periodically copied to $q_{\text{target}}$. Optimizing the temporal difference error (Eq. 8) is very common in reinforcement learning (Mnih et al., 2015; Van Hasselt et al., 2016; Liu et al., 2025b). Obviously, the pre-trained ARROW is not adapted to the error accumulation induced by the adaptive rollout policy $\pi_{\text{ARROW-P}}$. Although ARROW can be fine-tuned under $\pi_{\text{ARROW-P}}$ using Eq. 9, the resulting optimal policy will no longer be

$\pi_{\text{ARROW-P}}$, as the environment $\Psi$ changes due to the fine-tuning process.

$$\mathcal{L}_{q_{\text{target}}} = \mathop{\mathbb{E}}_{\Gamma \sim \mathfrak{S}(q_{\text{target}}, \Psi)} \left[ \frac{1}{|\Gamma| V H W} \sum_{\tau \in \Gamma} \sum_{v=1}^{V} \sum_{i=1}^{H} \sum_{j=1}^{W} w(v) Lat(i) (\hat{\Delta}_{\tau}^{vij} - \Delta_{\tau}^{vij})^2 \right], \qquad (9)$$

where the rollout trajectory $\Gamma$ is generated by the AR Scheduler $\mathfrak{S}$ through interaction with the environment, and $\Delta_{\tau}$ is the weather changes described in Section 2.1. The correct approach is to jointly optimize both the environment $\Psi$ and the value estimator $q_{\text{main}}$ as shown below:

$$\begin{cases} \arg\min_{q} \mathcal{L}_{\Psi^*}(q_{\text{main}}, q_{\text{target}}) \\ s.t., \ \Psi^* = \arg\min_{\Psi} \mathcal{L}_{q_{\text{target}}}(\Psi) \end{cases}, \quad \Gamma^* = \mathfrak{S}(q^*, \Psi^*). \qquad (10)$$

Here, $\Gamma^*$ denotes the trajectory generated by the optimal policy with learned state–action values $q^*$ and the environment $\Psi^*$ with adaptive trajectory fine-tuned ARROW. However, $\Psi$ and $q(s, a)$ influence each other, forming a bi-level optimization problem that makes Eq. 10 difficult to optimize jointly. To address this, we design an adaptive rollout fine-tuning algorithm that alternately optimizes these two objectives, allowing $\Psi$ and $q(s, a)$ to gradually converge toward optimality (see Appendix D for algorithmic details).

# 4 EXPERIMENTS

## 4.1 EXPERIMENT SETTINGS

**Datasets.** We evaluate performance using the WeatherBench1[1] benchmark, a subset of the ERA5 reanalysis dataset. For meteorological variables, we select five pressure-level variables: geopotential (Z), specific humidity (Q), temperature (T), and the U and V components of wind speed. Each variable is provided at 13 pressure levels (50, 100, 150, 200, 250, 300, 400, 500, 600, 700, 850, 925, and 1000 hPa). In addition, we include four surface-level variables: 2-meter temperature (T2m), 10-meter U and V wind components (U10 and V10), and total cloud cover (TCC). The spatial resolution is set to $1.40625°$, corresponding to a $128 \times 256$ grid. The temporal resolution is 6-hourly. The dataset spans 2008–2016 for training, 2017 for validation, and 2018 for testing. During evaluation, we focus on four surface variables: U10, V10, T2m, and TCC, and two pressure-level variables: Z500 and T850. Notably, Z500 reflects the mid-tropospheric dynamical structure, while T850 is closely linked to surface temperature; together, they capture key climatic drivers. U10, V10, and T2m are highly relevant to human activities, whereas TCC plays an important role in renewable energy applications (e.g., photovoltaic power generation).

**Baselines.** We evaluate ARROW against baselines from four categories: 1) Classical Methods: Climatology (Jung & Leutbecher, 2008) and the Integrated Forecast System (IFS) (Roberts et al., 2018). 2) Graph Neural Network-based Methods: Keisler (Keisler, 2022) and GraphCast (Lam et al., 2023). 3) Neural Operator-based Methods: FourcastNet (Pathak et al., 2022). 4) Transformer-based Methods: Pangu-weather (Bi et al., 2023) and Stormer (Nguyen et al., 2024). Details of all baselines are provided in Appendix E.

**Evaluation Metrics.** We conduct the evaluation based on the average of n-day forecasts, including four target lead times each day: 00:00 UTC, 06:00 UTC, 12:00 UTC, and 18:00 UTC. All forecasting methods are assessed using latitude-weighted Root Mean Square Error (RMSE) and Anomaly Correlation Coefficient (ACC) on de-normalized predictions (see Appendix F for more details). Since initial weather states in the IFS results are only available at 00:00 UTC and 12:00 UTC, IFS is included in the tables for reference only and excluded from overall comparison.

## 4.2 PERFORMANCE COMPARISON

**Overall Performance.** Table 4 compares the performance of ARROW against baseline models over a nine-day period (from Day 5 to Day 14) across six key meteorological variables. We use '↑' (and '↓') to indicate that larger (and smaller) values are better. ARROW consistently outperforms all data-driven baselines in both RMSE and ACC on all evaluated days. On average, ARROW achieves improvements of **9.3%** in RMSE and **10%** in ACC over five days across six atmospheric variables compared to the second-best data-driven models.

---

[1] https://github.com/pangeo-data/WeatherBench

Table 1: Overall prediction performance comparison. The best and second-best results are highlighted in **bold** and underline, respectively.

| Methods | | Climatology | IFS | | Keisler | | GraphCast | | Pangu-weather | | FourcastNet | | Stormer | | ARROW | |
|---|---|---|---|---|---|---|---|---|---|---|---|---|---|---|---|---|
| Variable | Lead Time | RMSE↓ | RMSE↓ | ACC↑ | RMSE↓ | ACC↑ | RMSE↓ | ACC↑ | RMSE↓ | ACC↑ | RMSE↓ | ACC↑ | RMSE↓ | ACC↑ | RMSE↓ | ACC↑ |
| T2m | 5-day | 2.74 | 1.83 | 0.76 | 3.20 | 0.46 | 1.84 | 0.72 | 2.37 | 0.59 | 2.39 | 0.59 | 1.76 | 0.78 | **1.66** | **0.80** |
|  | 7-day |  | 2.31 | 0.61 | 3.87 | 0.31 | 2.32 | 0.55 | 2.74 | 0.46 | 2.85 | 0.43 | 2.28 | 0.63 | **2.13** | **0.67** |
|  | 9-day |  | 2.73 | 0.45 | 4.47 | 0.21 | 2.70 | 0.40 | 3.05 | 0.34 | 3.17 | 0.31 | 2.72 | 0.47 | **2.50** | **0.52** |
|  | 14-day |  | - | - | 5.84 | 0.08 | 3.24 | 0.18 | 3.54 | 0.16 | 3.58 | 0.15 | 4.15 | 0.20 | **2.99** | **0.29** |
| U10 | 5-day | 3.96 | 2.89 | 0.72 | 4.49 | 0.37 | 3.02 | 0.65 | 3.50 | 0.54 | 3.91 | 0.47 | 3.03 | 0.70 | **2.84** | **0.73** |
|  | 7-day |  | 3.79 | 0.52 | 5.21 | 0.21 | 3.77 | 0.44 | 4.05 | 0.37 | 4.51 | 0.29 | 3.86 | 0.50 | **3.59** | **0.54** |
|  | 9-day |  | 4.44 | 0.34 | 5.74 | 0.12 | 4.26 | 0.28 | 4.42 | 0.24 | 4.85 | 0.19 | 4.39 | 0.33 | **4.09** | **0.37** |
|  | 14-day |  | - | - | 6.90 | 0.03 | 4.77 | 0.09 | 4.86 | 0.09 | 5.24 | 0.07 | 5.01 | 0.10 | **4.54** | **0.15** |
| V10 | 5-day | 4.02 | 2.99 | 0.71 | 4.72 | 0.34 | 3.15 | 0.64 | 3.58 | 0.53 | 4.06 | 0.45 | 3.14 | 0.68 | **2.94** | **0.72** |
|  | 7-day |  | 3.95 | 0.49 | 5.52 | 0.18 | 3.96 | 0.42 | 4.18 | 0.35 | 4.71 | 0.26 | 4.02 | 0.48 | **3.74** | **0.52** |
|  | 9-day |  | 4.63 | 0.30 | 6.11 | 0.09 | 4.48 | 0.25 | 4.60 | 0.20 | 5.05 | 0.15 | 4.59 | 0.29 | **4.25** | **0.34** |
|  | 14-day |  | - | - | 7.38 | 0.01 | 4.99 | 0.07 | 5.06 | 0.05 | 5.42 | 0.04 | 5.15 | 0.07 | **4.71** | **0.11** |
| TCC | 5-day | 0.31 | - | - | 0.45 | 0.14 | 0.30 | 0.47 | 0.34 | 0.26 | 0.32 | 0.31 | 0.29 | 0.47 | **0.28** | **0.51** |
|  | 7-day |  | - | - | 0.51 | 0.08 | 0.32 | 0.35 | 0.35 | 0.20 | 0.35 | 0.22 | 0.32 | 0.34 | **0.31** | **0.38** |
|  | 9-day |  | - | - | 0.56 | 0.05 | 0.37 | 0.23 | 0.36 | 0.16 | 0.36 | 0.16 | 0.34 | 0.24 | **0.32** | **0.28** |
|  | 14-day |  | - | - | 0.72 | 0.02 | 0.42 | 0.09 | 0.38 | 0.09 | 0.37 | 0.09 | 0.39 | 0.10 | **0.35** | **0.15** |
| Z500 | 5-day | 807.15 | **358.96** | **0.90** | 638.49 | 0.67 | 461.78 | 0.83 | 510.13 | 0.79 | 617.48 | 0.69 | 394.11 | 0.88 | 370.75 | **0.90** |
|  | 7-day |  | 576.46 | 0.74 | 821.51 | 0.45 | 652.47 | 0.65 | 704.35 | 0.59 | 800.68 | 0.48 | 604.04 | 0.71 | **565.20** | **0.74** |
|  | 9-day |  | 765.21 | 0.54 | 941.20 | 0.29 | 801.25 | 0.46 | 843.45 | 0.42 | 921.87 | 0.32 | 764.03 | 0.52 | **715.46** | **0.56** |
|  | 14-day |  | - | - | 1,139.31 | 0.06 | 987.24 | 0.15 | 1,027.68 | 0.14 | 1,063.54 | 0.11 | 996.46 | 0.19 | **887.37** | **0.25** |
| T850 | 5-day | 3.45 | **2.04** | **0.82** | 3.44 | 0.51 | 2.29 | 0.76 | 2.62 | 0.69 | 2.99 | 0.60 | 2.25 | 0.78 | 2.06 | **0.82** |
|  | 7-day |  | 2.86 | 0.64 | 4.19 | 0.31 | 3.01 | 0.57 | 3.32 | 0.50 | 3.66 | 0.41 | 3.05 | 0.60 | **2.77** | **0.65** |
|  | 9-day |  | 3.53 | 0.46 | 4.75 | 0.18 | 3.58 | 0.39 | 3.83 | 0.34 | 4.11 | 0.26 | 3.67 | 0.42 | **3.32** | **0.48** |
|  | 14-day |  | - | - | 5.91 | 0.01 | 4.28 | 0.13 | 4.47 | 0.12 | 4.62 | 0.09 | 4.71 | 0.13 | **3.91** | **0.20** |

*Note:* Climatology is derived from historical averages and therefore yields a constant RMSE across all target lead times throughout the year. By definition, ACC is not applicable to Climatology. The IFS results exclude forecasts beyond 10 days as well as the TCC variable. Missing results in both cases are denoted by "–".

From Table 4 we can also observe the following: 1) Deep learning methods exhibit strong performance in short-term forecasting (e.g., 5-day), even surpassing IFS in some cases, indicating their potential. 2) Keisler performs poorly overall, especially on 9-day and 14-day forecasts, where its predictions are nearly unusable. In contrast, GraphCast demonstrates superior performance through various technical advancements, such as the multi-scale graphs. It highlights the potential of graph neural networks in weather forecasting. 3) Neural operators have demonstrated the feasibility of operator learning in weather forecasting, but they still lag behind the latest data-driven models. 4) Transformer-based approaches achieve competitive forecasting performance by leveraging computer vision techniques, such as the Swin Transformer. Overall, these results highlight ARROW's effectiveness in capturing precise spatiotemporal correlations within atmospheric dynamics while adaptively scheduling rollout trajectories during autoregressive prediction—demonstrating clear advantages in medium-range weather forecasting.

## 4.3 Ablation Studies

**Effect of Ring Positional Encoding.** To validate the effectiveness of RPE in ARROW, we replace it with the 2D Positional Encoding from (Dosovitskiy et al., 2020) and make it learnable, as shown in Table 2 (w/o RPE). Since 2D Positional Encoding is originally designed for images rather than the Earth, both RMSE and ACC deteriorate in 72h lead time.

Table 2: Effect of RPE, S&P MoE and two auxiliary losses. All methods are evaluated at the 72-hour lead time using the same naive rollout strategy (Nguyen et al., 2024).

| Methods | T2m-72h | | U10-72h | | V10-72h | |
|---|---|---|---|---|---|---|
|  | RMSE↓ | ACC↑ | RMSE↓ | ACC↑ | RMSE↓ | ACC↑ |
| w/o RPE | 1.12 | **0.91** | 1.76 | 0.90 | 1.82 | 0.90 |
| w/o S&P MoE | 1.13 | **0.91** | 1.77 | 0.90 | 1.81 | 0.90 |
| w/o aux-loss$_1$ | 1.15 | 0.89 | 1.83 | 0.88 | 1.92 | 0.88 |
| w/o aux-loss$_2$ | 1.12 | 0.90 | 1.82 | 0.88 | 1.85 | 0.89 |
| ARROW-Pretrain | **1.09** | **0.91** | **1.71** | **0.91** | **1.77** | **0.91** |

**Effect of Shared-Private Mixture-of-Experts.** To validate the capability of S&P MoE in capturing weather variations across different intervals, we replace S&P MoE with a FFN, as shown in Table 2 (w/o S&P MoE). The results indicate that removing the ability to capture interval-specific

atmospheric variations leads to performance degradation. This underscores the importance of both shared and specific presentations across multi-scale temporal dependencies in atmospheric dynamics.

**Effect of two auxiliary losses.** To further demonstrate the effectiveness of the two auxiliary losses for S&P MoE, we design experiments that remove each loss individually: w/o aux-loss$_1$ and w/o aux-loss$_2$, as shown in Table 2. When aux-loss$_1$ is moved, the model focuses solely on load balancing across experts, preventing individual private experts from capturing distinct features. In contrast, removing aux-loss$_2$ eliminates load balancing, causing the model to heavily rely on only a few experts.

**Effect of Adaptive Rollout Strategy.** To assess the impact of the rollout strategy, we compare ARROW (**Adaptive** in Fig. 3) with the three strategies: a) **Random**, which selects time intervals randomly during autoregression, b) **Naive**, which repeats 23 times with time interval 6h, and c) **Greedy**, adopted by Pangu-Weather (e.g., $138h = 24h \times 5 + 12h + 6h$). As shown in Fig. 3, the results for T2m and T850 at the 138-hour lead time suggest that the adaptive strategy, conditioned on the current weather state, is crucial for robust forecasting. Although the greedy strategy alleviates error accumulation, it lacks the ability to capture

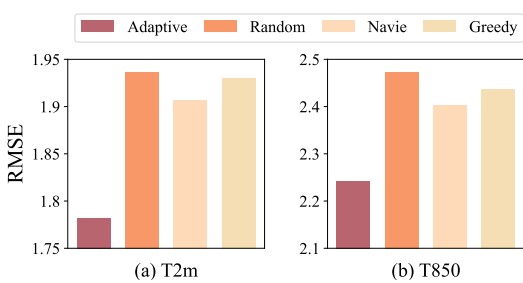

Figure 3: Effect of Adaptive Rollout Scheduler on T2m and T850 at a 138-hour lead time.

fine-grained atmospheric variations and therefore performs worse than the naive strategy. Most importantly, the adaptive strategy outperforms the random strategy, demonstrating that the AR scheduler successfully estimates meaningful state–action values within the weather forecasting environment.

## 4.4 CASE STUDIES

Two representative examples are analyzed to demonstrate ARROW's strong forecasting capability and its potential for downstream applications. To highlight forecasting performance under extreme weather events, we visualize the Siberian cold wave (as shown in Fig. 4) that occurred from January 22 to January 27, 2018, and its impact on East and Central China (purple circle) as well as several Central Asian countries (black circle). Based on the initial weather state, ARROW successfully predicted the trajectory of the cold wave and accurately captured the temperature variations in the marked regions. Notably, the red spot within the purple circle is shielded by topographic factors and remains unaffected by the cold wave. This detail is precisely predicted by ARROW, demonstrating its potential for air quality (Liang et al., 2023; Tian et al., 2025) and traffic flow (Wu et al., 2021; 2023; 2024) prediction tasks in which temperature serves as an important auxiliary covariate.

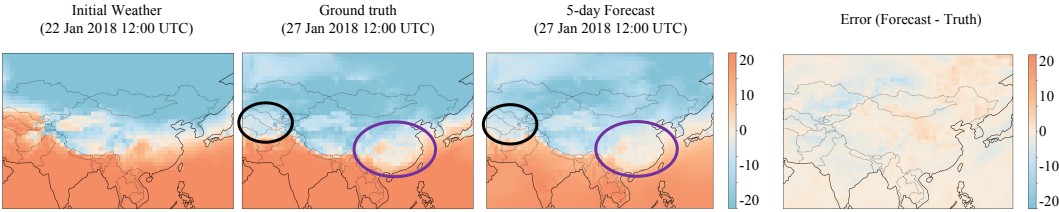

Figure 4: Visualization of T2m during the Siberian cold wave (unit: °C).

Next, we visualize changes in cloud cover from May 10 to May 11, 2018. Cloud cover prediction plays a crucial role in decision-making for photovoltaic power scheduling. As shown in Fig. 5, ARROW accurately predicted global cloud cover one day in advance. In particular, as indicated by the two black circles, ARROW successfully identified regions of low cloud cover, providing decision support for adjusting photovoltaic power generation in these areas.

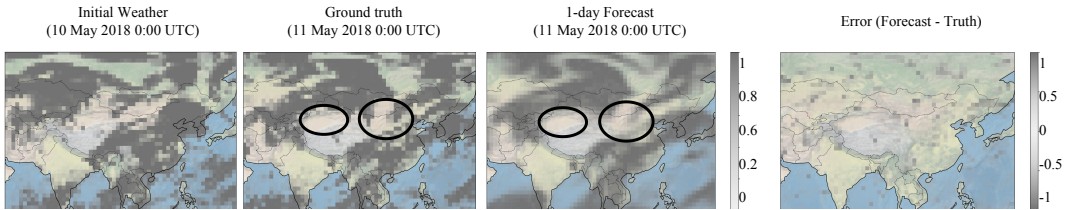

Figure 5: Visualization of TCC in China (unitless).

To demonstrate ARROW's capability to capture fine-grained spatiotemporal dependencies in other meteorological variables, we provide additional comparative case studies (see Appendix J).

## 5  CONCLUSION

In this paper, we present ARROW, an adaptive rollout and routing method for global weather forecasting. ARROW consists of two key components: a Multi-Interval Forecasting Model (MIFM) and an Adaptive Rollout scheduler (AR scheduler). Leveraging Ring Positional Encoding and a Shared-Private Mixture-of-Experts, the pre-trained MIFM effectively captures spatial and multi-scale temporal dependencies in atmospheric dynamics. The AR Scheduler is designed to select appropriate autoregressive rollout strategies. By alternately optimizing the temporal-difference loss and performing multi-step fine-tuning on the pre-trained model, ARROW effectively suppresses cumulative errors under the learned policy. In the future, we plan to explore reinforcement learning for local weather forecasting and further investigate physics-guided approaches in this domain.

## ACKNOWLEDGEMENT

This work was partially supported by the National Natural Science Foundation of China (62372179). In addition, we thank Prof. S. Yang and Dr. S. Liu for their helpful discussions during this work.

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

APPENDIX

## A NOTATION

Table 3: Summary of Notations used in the paper

| Symbol | Description |
|---|---|
| $\delta$ | Time interval |
| $P$ | Patch size in the tokenization |
| $V$ | Number of Atmospheric variables |
| $H$ | Latitude in the grid |
| $W$ | Longitude in the grid |
| $h$ | Token resolution (latitude), $h = (H/P)$ |
| $w$ | Token resolution (longitude), $w = (W/P)$ |
| $X_0$ | Initial weather state |
| $X_\tau$ | Current Weather state in time $\tau$ |
| $\Psi$ | Weather forecasting environment with pre-trained ARROW and weather dataset |
| $\Gamma$ | Rollout trajectory to $X_T$, $\Gamma = \mathfrak{S}(q, \Psi) = \{\tau_1, \tau_2, ..., \tau_{|\Gamma|}\}$ |
| $G_\Gamma$ | Return of the trajectory $\Gamma$, $G_\Gamma = \sum_{\tau \in \Gamma} r_\tau + \omega$ |
| $\mathfrak{S}$ | Adaptive Rollout Scheduler (AR Scheduler) |
| $Z^{(n)}$ | Weather embedding used as the input to the $n$-th Arch Block |
| $\bar{Z}^{(n)}$ | Input representation to the S&P MoE in the $n$-th Arch Block |
| $t$ | The step in the trajectory $\Gamma$ |
| $\tau_t$ | The timestamp at the $t$-th in the trajectory $\Gamma$, $\tau_t = \Gamma[t]$ |

## B THE DETAILS OF RING POSITIONAL ENCODING

For clarity, we first introduce the formulation of 1D Ring Positional Encoding, and then extend it to 2D Ring Positional Encoding.

**1D Ring Positional Encoding.** Suppose there is a sequence $X \in \mathbb{R}^{W \times D} = \{x_1, x_2, \ldots, x_S\}$ that forms a circular structure. The distance between any two positions on the sequence satisfies the following property:

$$\forall\, a, b \in \{1, 2, \ldots, W\}, \quad \text{dis}_{a,b} = \min\{|a - b|,\ W - |a - b|\}, \tag{11}$$

where $\text{dis}_{a,b}$ denotes the circular distance between positions $a$ and $b$ on a periodic sequence of length $W$, defined as the minimum of the direct distance and the wrap-around distance. Following the sine/cosine positional encoding (Vaswani et al., 2017; Xu et al., 2025), RPE should be in this form:

$$\text{RPE}_{k,2i} = sin(\frac{k \cdot 2\pi}{W} \cdot i) \cdot \frac{S}{2i}, \quad \text{RPE}_{k,2i+1} = cos(\frac{k \cdot 2\pi}{W} \cdot i) \cdot \frac{S}{2i}, \tag{12}$$

where $i = 1, \ldots, \frac{D}{2}$. By computing the dot-product similarity, it is straightforward to obtain

$$\langle P_m, P_n \rangle = \sum_i^{\frac{d}{2}} cos(\frac{m - n}{L} \cdot 2\pi i) \cdot (\frac{L}{2i})^2.$$

It can be proven to satisfy the property described in Eq. 11.

**2D Longitude Ring Positional Encoding.** Inspired by the 2D sine/cosine positional encoding (Dosovitskiy et al., 2020; Heo et al., 2024), the 1D RPE can be naturally extended to 2D. In practice, the input vector is split into two halves: one half is assigned the one-dimensional circular positional encoding along the longitude direction, and the other half is assigned the one-dimensional

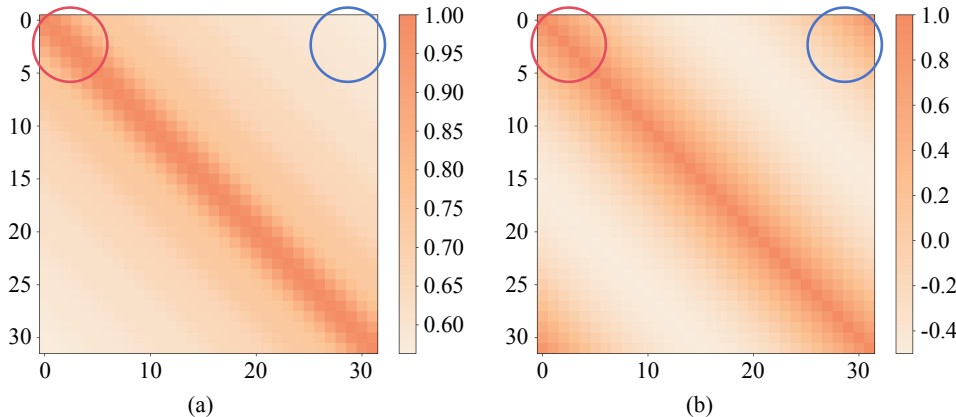

Figure 6: Similarity matrix of normal positional encoding and ring positional encoding.

positional encoding along the latitude direction. Therefore, for any location $k : (p_k^x, p_k^y)$ on an $h \times w$ grid, its positional encoding is given by:

$$\text{RPE}_{k,4i} = sin(\frac{p_k^x}{w} \cdot 2\pi i) \cdot \frac{w}{4i}, \ \ \text{RPE}_{k,4i+1} = cos(\frac{p_k^x}{w} \cdot 2\pi i) \cdot \frac{w}{4i}, \tag{13}$$

$$\text{RPE}_{k,4i+2} = sin(\frac{p_k^y}{w} \cdot 2\pi i) \cdot \frac{w}{4i}, \ \ \text{RPE}_{k,4i+3} = cos(\frac{p_k^y}{w} \cdot 2\pi i) \cdot \frac{w}{4i}, \tag{14}$$

where $i = 1, ..., \frac{D}{4}$. Note that properties of Eq. 12 are not applied on $4i$ and $4i + 1$ because the latitude has not the ring property after equirectangular projection.

**Differences between Ring Positional Encoding and Conventional Positional Encoding.** We visually contrast RPE with conventional positional encoding in Fig. 6. The equation of conventional positional encoding is shown as follows:

$$\begin{cases} P_{k,2i} = sin(k \cdot 10000^{-\frac{2i}{d}}), \\ P_{k,2i+1} = cos(k \cdot 10000^{-\frac{2i}{d}}). \end{cases} \tag{15}$$

As demonstrated: Conventional 1D positional encoding exhibits high similarity values (red circle in Fig. 6 (a)) between positions near the left endpoint, while showing low similarity (blue circle in Fig. 6 (a)) between left-end and right-end regions. This fails to recognize endpoint adjacency, which contradicts Earth's longitudinal continuity where 180°W and 180°E are identical. Conversely, RPE correctly maintains high similarity (blue circle in Fig. 6 (b)) between endpoints, accurately capturing cyclic spatial relationships essential for global meteorological data processing.

## C  OPTIMIZATION LOSS

Clearly, a higher value of aux-loss$_1$ and a lower value of aux-loss$_2$ are desirable. Therefore, the overall auxiliary loss function is defined as:

$$\mathcal{L}_{\text{aux}} = -\text{aux-loss}_1 + \alpha \cdot \text{aux-loss}_2, \tag{16}$$

where $\alpha$ is a weighting hyperparameter. In addition, to train ARROW to forecast atmospheric dynamics at random time intervals, conditioned on $\delta$, we introduce a randomized dynamics forecasting loss $\mathcal{L}_\delta$ as follows:

$$\mathcal{L}_\delta = \underset{\delta \sim P(\delta)}{\mathbb{E}} \left[ \frac{1}{VHW} \sum_{v=1}^{V} \sum_{i=1}^{H} \sum_{j=1}^{W} w(v) L(i) (\hat{\Delta}_\delta^{vij} - \Delta_\delta^{vij}) \right]. \tag{17}$$

Here, $w(v)$ denotes the variable-specific loss weight (Lam et al., 2023; Nguyen et al., 2024), $L(i)$ is the latitude weighting factor, and $\hat{\Delta}_\delta$ represents the predicted dynamics corresponding to the time interval. Finally, the overall loss function used during the pre-training stage is summarized as:

$$\mathcal{L}_{\text{pre-train}} = \mathcal{L}_\delta + \mathcal{L}_{\text{aux}}. \tag{18}$$

# D The algorithm of adaptive rollout fine-tuning

## D.1 Algorithm details

Considering the bi-level optimization in Eq. 10, we adopt the following algorithm to alternately optimize $q$ and $\Psi$.

---

**Algorithm 1:** Adaptive Rollout Fine-tuning

---

**Input:** Weather forecasting environment $\Psi$ with Pre-trained ARROW and weather dataset, AR Scheduler $\mathfrak{S}$, replay buffer $\mathcal{B}$, penalty coefficient $\omega$, max optimizing steps $T_{\max}$, alternating optimization rate $C$

**Output:** Fine-tuned ARROW, AR Scheduler

1   Initialize Weather Forecasting Environment $\Psi$;
2   Initialize Main DQN $q_{\mathrm{main}}(s, a)$ and Target DQN $q_{\mathrm{target}}(s, a)$;
3   Initialize replay buffer $\mathcal{B}$;
4   **for** *each epoch* **do**
5      Interact agent with environment and store $(s, a, r, s')$ into $\mathcal{B}$;
6      Refresh $\mathcal{B}$ with $q_{\mathrm{target}}$;
7      **for** *each iteration* **do**
8          Uniformly sample a mini-batch from $\mathcal{B}$;
9          Optimizing the Eq. 8 as follows:
10         **for** *each transition* $(s, a, r, s')$ *in the mini-batch* **do**
11             Compute target value:
12               $y_{\mathrm{target}} = r + \gamma \cdot \max\limits_{a \in \mathcal{A}(s')} q_{\mathrm{target}}(s', a)$;
13             Update $q_{\mathrm{main}}$ by minimizing Temporal difference loss:
14               $\mathcal{L}_\Psi = (y_{\mathrm{target}} - q_{\mathrm{main}}(s, a))^2$;
15      **if** *global step* $\mathrm{mod}\ C = 0$ **then**
16          Copy parameters: $q_{\mathrm{target}} \leftarrow q_{\mathrm{main}}$;
17          Optimizing the Eq. 9 as follows:
18          AR Scheduler generates rollout trajectory $\Gamma$ based on initial weather state:
19             $\Gamma = \mathfrak{S}(q_{\mathrm{target}, \Psi})$, like $\Gamma = \{6h, 18h, 42h, \dots, 120h\}$ ;
20          Fine-tuning ARROW's prediction head with rollout loss on the trajectory $\Gamma$:
21             Stop the gradient of $\{\hat{\Delta}_\tau\}_\Gamma$ for steps where $T_{\max} < |\Gamma|$.
22             $\mathcal{L}_{q_{\mathrm{target}}} = \sum_{\tau \in \Gamma} (\Delta_\tau - \hat{\Delta}_\tau)^2$;

---

## D.2 Algorithm stability and convergence

Given that Adaptive Rollout Fine-tuning (Algorithm 1) is an alternative optimization of Eq. 8 and Eq. 9, it is essential to verify the numerical stability of the algorithm. Therefore, we provide the training curves of the multi-step fine-tuning loss and temporal difference loss (Fig. 7), both exhibiting clear convergence. Additionally, we also present the trajectory return, which shows an upward trend consistent with the convergence of the two losses. Since the reward is defined as the negative RMSE (as described in Sec. 3.3's Reward Function), the objective of the AR Scheduler is to make the trajectory return approach zero. These results indicate that the AR Scheduler learns more effective rollout trajectories for weather prediction compared to random initialization.

## E Baseline Details

### E.1 IFS HRES

The IFS HRES (Roberts et al., 2018) (High Resolution Forecast System) is ECMWF's flagship deterministic forecasting model, developed based on its core Integrated Forecasting System (IFS). It represents one of the world's most advanced global numerical weather prediction technologies for

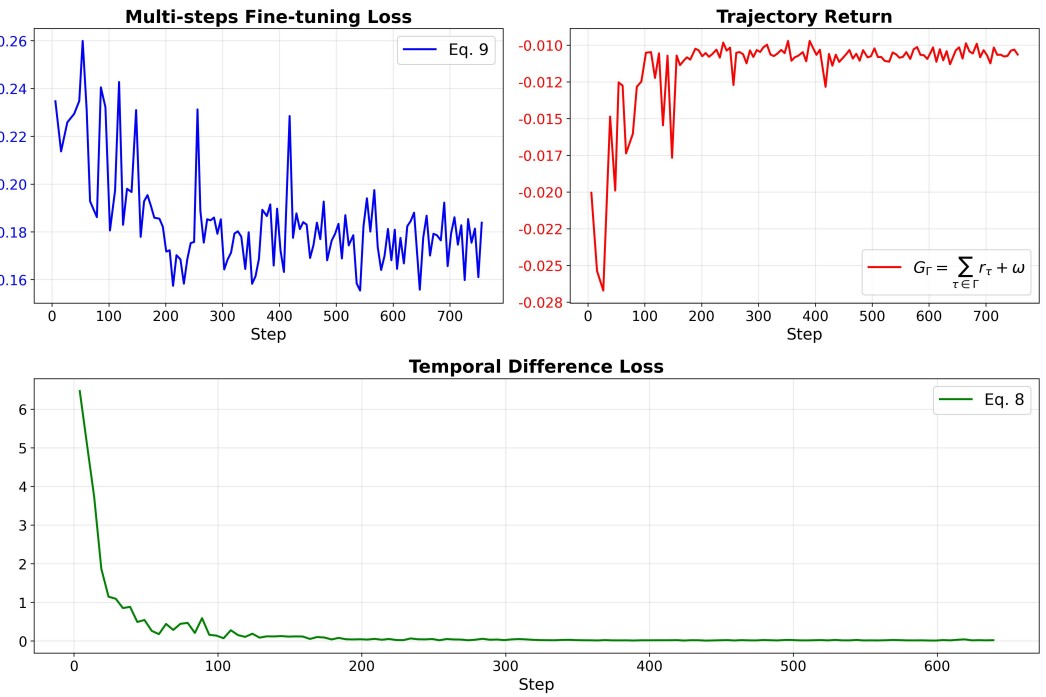

Figure 7: The training curves of the temporal difference loss (Eq. 8), multi-step fine-tuning loss (Eq. 9), and trajectory return collectively demonstrate that the AR Scheduler achieves convergence.

short-to-medium-range forecasting. By assimilating massive observational data—including satellite, surface station, and radiosonde observations—through advanced four-dimensional variational data assimilation (4D-Var), the model drives high-fidelity physical equations to generate global weather forecasts.

### E.2    STORMER

Stormer (Nguyen et al., 2024) is a global weather prediction model based on the Transformer architecture (derived from DiT). Its training process consists of two phases: initial pre-training via single-step prediction to establish short-term forecasting capabilities, followed by critical multi-step fine-tuning that focuses on 4-step and 8-step prediction performance (including ablation studies validating its significance). During inference, Stormer adopts an innovative composite prediction strategy, generating weather forecasts by averaging outputs across multiple time intervals, thereby substantially improving forecast stability and accuracy. For example, the results of three 24-hour, six 12-hour, and twelve 6-hour predictions are ensembled to produce the final 72h forecast.

### E.3    KEISLER GRAPH NEURAL NETWOR

The Keisler Graph Neural Network (Keisler, 2022) is an innovative global weather prediction model that pioneers the application of graph neural networks (GNNs) in this domain. Its core architecture employs a bipartite graph neural network (featuring encoder and decoder components) coupled with a specialized GNN processor.

### E.4    GRAPHCAST

The GraphCast (Lam et al., 2023) is a global weather forecasting model that leverages multi-scale graph to model the evolution of atmospheric variables over time. It represents the Earth as a spherical mesh of nodes connected by edges based on geodesic proximity, and employs a message-passing

GNN architecture to learn spatial and temporal dependencies. Similar with Keisler, GraphCast employs a bipartite graph to construct the connection between the graph(mesh) and grid. By encoding a sequence of past meteorological states and predicting the next state in an autoregressive manner, GraphCast achieves fast and accurate medium-range forecasts. It significantly outperforms traditional numerical weather prediction models and earlier deep learning approaches in both accuracy and inference speed.

### E.5 PANGU-WEATHER

Pangu-Weather (Bi et al., 2023) is a model specialized in global high-resolution weather prediction. Built upon a 3D Earth-Specific Transformer architecture—tailored to process the three-dimensional spatial characteristics of Earth's meteorological data (longitude, latitude, altitude)—its core training strategy involves concurrently training three independent models. During inference, the system employs a greedy strategy for any target forecast horizon: it iteratively selects and utilizes the model with the largest available timestep (24h, 12h or 6h) to efficiently achieve long-term forecasting.

### E.6 FORECASTNET

FourCastNet (Pathak et al., 2022) is an advanced AI model specifically designed for global high-resolution weather forecasting. Built on a Transformer architecture integrated with Adaptive Fourier Neural Operators (AFNO), it efficiently processes meteorological data across a global $128 \times 256$ grid (WeatherBench dataset).

## F METRICS

### F.1 ROOT MEAN SQUARED ERROR (RMSE)

The RMSE is calculated as follows:

$$\text{RMSE} = \frac{1}{T} \sum_{t=1}^{T} \sqrt{\frac{1}{V \times H \times W} \sum_{v=1}^{V} \sum_{i=1}^{H} \sum_{j=1}^{W} L(i) \left( \hat{X}_{t,v,i,j} - X_{t,v,i,j} \right)^2}, \tag{19}$$

where $T$ denotes the number of time steps, $V$ represents the number of variables, $H$ and $W$ correspond to the height and width of grids, respectively. $L(\cdot)$ is the latitude weighting factor (decreasing weight with increasing latitude). Here, $\hat{X}_{t,v,i,j}$ and $X_{t,v,i,j}$ correspond to ARROW's forecast and the ground truth for variable $V$ at time $t$ and grid point $(i, j)$.

### F.2 ANOMALY CORRELATION COEFFICIENT (ACC)

The ACC computes the Pearson correlation coefficient between climatology-based anomalies $\hat{X}'_{t,v,i,j} = \hat{X}_{t,v,i,j} - C_{t,v,i,j}$ and $X'_{t,v,i,j} = X_{t,v,i,j} - C_{t,v,i,j}$, where $C_{t,v,i,j}$ represents the climatology for variable $v$ at grid point $(i, j)$ and time step $t$. The ACC is defined as:

$$\text{ACC} = corr(\hat{X}', X') = \frac{1}{T} \sum_{t=1}^{T} \frac{\sum_{i,j} L(i) \hat{X}'_{t,v,i,j} X'_{t,v,i,j}}{\sqrt{\sum_{i,j} L(i) \hat{X}'^2_{t,v,i,j} \sum_{i,j} L(i) X'^2_{t,v,i,j}}}. \tag{20}$$

This metric evaluates whether predicted anomalies (deviations from climatology) align spatially with true anomalies. Higher ACC values (closer to 1) indicate superior model capability in capturing anomalous patterns, while lower values (near 0) reflect weaker performance.

## G   EXPERIMENTS DETAILS

### G.1   SOFTWARE AND HARDWARE ENVIRONMENT

All experiments are conducted using PyTorch 2.1.0 and PyTorch Lightning 2.1.1, executed on 8 NVIDIA GeForce A800 GPUs (80 GB). We leverage mixed-precision training and the Distributed Data Parallel (DDP) strategy. The software environment includes CUDA 11.8 and Python 3.11, running on Ubuntu 22.04.

### G.2   IMPLEMENTS DETAILS

**Pre-training.** In ARROW, the patch size is set to 4 for tokenization. The core architecture comprises 16 Transformer layers, each with a hidden size of 1024, and the attention mechanism employs 16 heads. The S&P MoE module includes 1 shared expert and 9 private experts, with 3 experts selected for each forward pass. The time intervals are configured as [6h, 12h, 24h], enabling ARROW to perform multi-interval forecasting and allowing the AR Scheduler to select temporal intervals for trajectory construction. During training, the model is trained with a batch size of 16 and an initial learning rate of 0.0005, which follows a cosine decay schedule after a 5-epoch warm-up phase. We adopt the AdamW optimizer with $\beta_1 = 0.9$, $\beta_2 = 0.95$, and a weight decay of 1e-5 to optimize the model parameters. We apply an early stopping mechanism with a patience of 10 epochs for optimization.

**Fine-tuning.** In the AR Scheduler, the replay buffer size is set to 9000, with a refresh size of 2000 at the end of each epoch. The agent learning rate and multi-step fine-tuning learning rate are set to 0.0002 and 1e-7, respectively. The alternating optimization interval is set to 10 steps. The target network $q_{\text{target}}$ is synchronized with the main network $q_{\text{main}}$ every 10 steps. The maximum number of optimization steps for multi-step fine-tuning, denoted as $T_{\text{max}}$, is set to 10 to adapt to error accumulation under the learned rollout strategy.

## H   ANALYSIS OF S&P MOE

To evaluate whether the routing behavior achieves both individual characteristics and global balance, we visualize the expert selection distribution across different time intervals within one Arch Block of ARROW (left side of Fig. 2). Three key phenomena can be observed from Fig. 8: (1) the overall load of S&P MoE is balanced; (2) no expert is simultaneously responsible for all three intervals, indicating that the shared expert is sufficient to capture the commonalities across intervals; (3) each private expert predominantly handles a specific interval while still being involved in a small number of other intervals. These findings support the design goals of S&P MoE: (i) capturing global

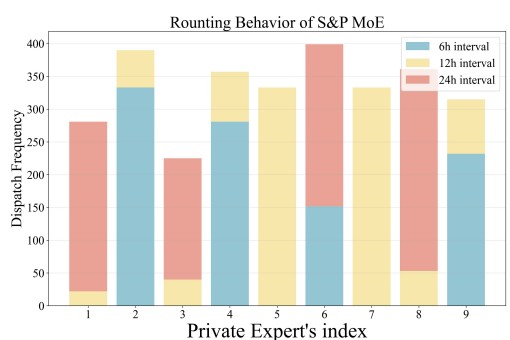

Figure 8: The routing behavior of S&P MoE.

commonalities shared across all time intervals, (ii) learning distinct features specific to each time interval, and (iii) representing partial commonalities shared by subsets of intervals (as discussed in Sec. 3.2.2).

## I   ANALYSIS OF AR SCHEDULER

In this section, we analyze the behavior of the AR Scheduler during inference on the 2018 test dataset and provide an in-depth analysis of the learned adaptive rollout strategy.

We first present the distribution of selected time intervals when making decisions for medium-range (120h) and long-range (354h) forecasts in Fig. 9. Notably, the decision trajectories for 120h forecasts rely more frequently on the 6-hour interval than those for 354h. This is because the trajectory length for 120h forecasts is generally shorter (see Fig. 10), allowing finer 6-hour intervals to capture detailed atmospheric dynamics without incurring significant accumulated error. In contrast, longer 354h trajectories cannot afford to use such short intervals due to greater potential for error accumulation. Thus, the AR Scheduler's interval selection behavior reflects a reasonable trade-off between temporal resolution and error accumulation.

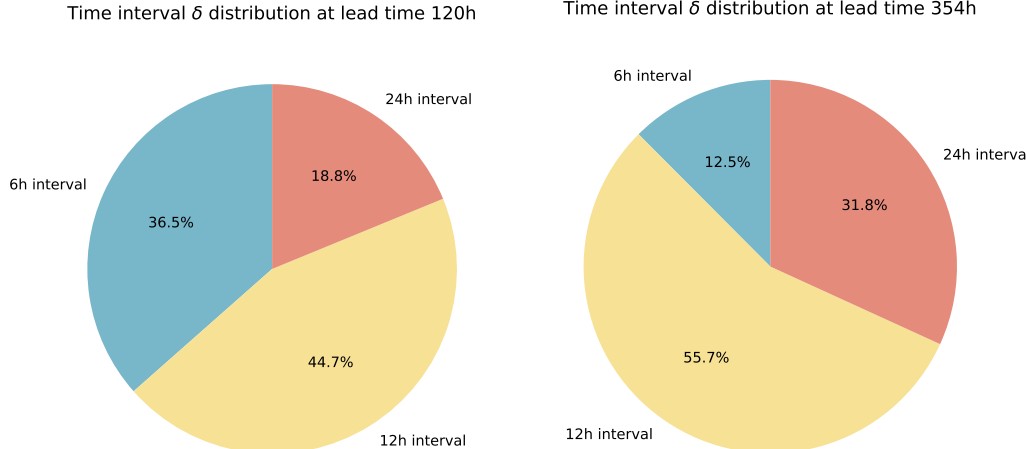

Figure 9: Decision interval distribution throughout 2018.

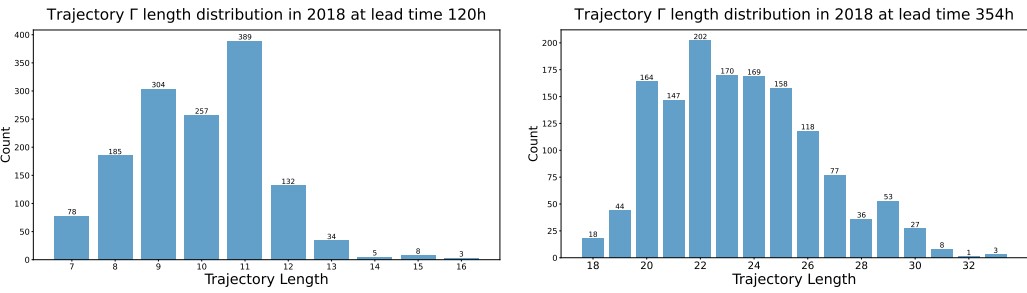

Figure 10: Trajectory lengths for 120h and 354h lead times throughout 2018.

Then, we further examine the AR Scheduler's decision trajectories for the 120h and 354h forecasts to investigate the rationale behind its interval choice under specific weather conditions. In Fig. 11, we present the weekly distribution of different intervals throughout 2018. Interestingly, the 6h interval shows a similar pattern in both the 120h and 354h forecasts, suggesting that the AR Scheduler considers fine-grained atmospheric dynamics to be equally important during certain periods, regardless of the lead time. Specifically, the black circles in Fig. 11 highlight periods of high frequency in selecting 6h interval.

To validate these decisions, we plot the temporal changes in geopotential in the 850hPa (Z850) and 1000hPa pressure level (Z1000) at each time step. As shown in Fig. 12, the time points highlighted with black circles closely align with those in Fig. 11. These periods exhibit clear signs of regime shifts in geopotential's variation height, characterized by pronounced curvature and turning points. Such transitions typically correspond to significant changes in global low-pressure systems and are often associated with severe weather phenomena, including strong winds, precipitation, and convection. The identified transition period, from late August to early September, marks a seasonal shift:

from summer to autumn in the Northern Hemisphere and from winter to spring in the Southern Hemisphere. This period is also associated with heightened tropical activity worldwide.

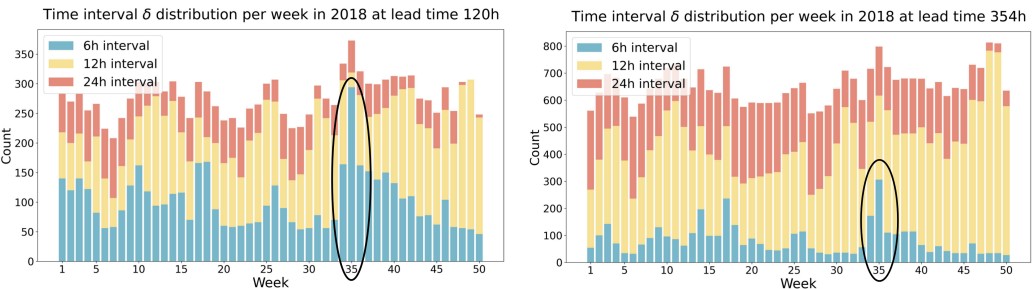

Figure 11: The weekly interval distribution in AR scheduler decision trajectories throughout 2018.

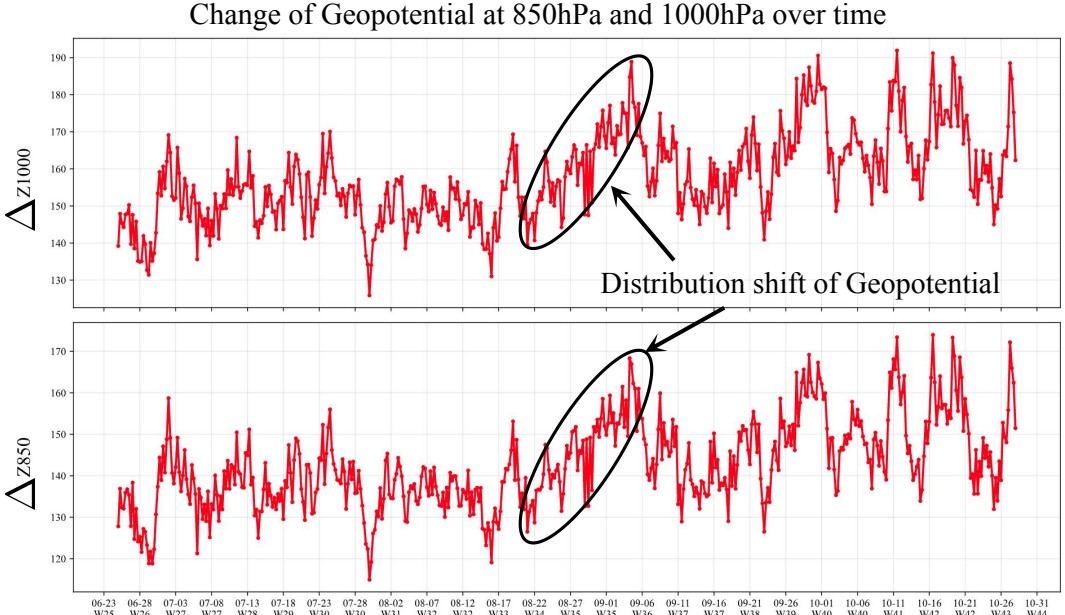

Figure 12: The changes of geopotenital at 850hPa and 1000hPa pressure level at each time step.

To further verify the meteorological context from August 25 to September 5, 2018, we retrieved historical weather reports from that period:

- On August 27, 2018, Hurricane Lane rapidly weakened into a tropical storm as it approached the Hawaiian Islands.
- Typhoon Jebi, which formed on August 26 and was officially named on August 27, made landfall in Japan's Kansai region on September 4. It became one of the most destructive typhoons to impact Japan in 2018.
- Tropical Storm Gordon developed around August 30, affecting regions such as Florida, the Gulf Coast, Arkansas, and Mississippi. It brought heavy rainfall, toppled trees, and caused widespread power outages.

These reports confirm the presence of multiple extreme weather events during that period. Since such events demand more fine-grained modeling to accurately capture atmospheric dynamics, the AR scheduler's frequent use of 6-hour intervals demonstrates its ability to correlate trajectory decisions with real-world weather patterns.

## J CASE STUDIES ON PREDICTION COMPARISONS

Fig. 13 presents a qualitative comparison of global weather forecasts at a 234-hour lead time across several models. Among them, ARROW demonstrates outstanding predictive accuracy and spatial coherence across all variables, closely matching the Ground Truth. It captures fine-grained patterns in T2m, U10, V10, Z500), and T850 with minimal artifacts and excellent global consistency. In contrast, Pangu-weather and FourCastNet exhibit noticeable spatial noise and over-smoothed patterns, especially over oceanic and polar regions, leading to reduced fidelity in key meteorological features. Stormer shows even more pronounced distortions and patchy inconsistencies, particularly in wind fields and geopotential contours, which compromises forecast reliability. Overall, ARROW not only recovers finer structures but also achieves superior global balance, making it reliable for long-term weather forecasts.

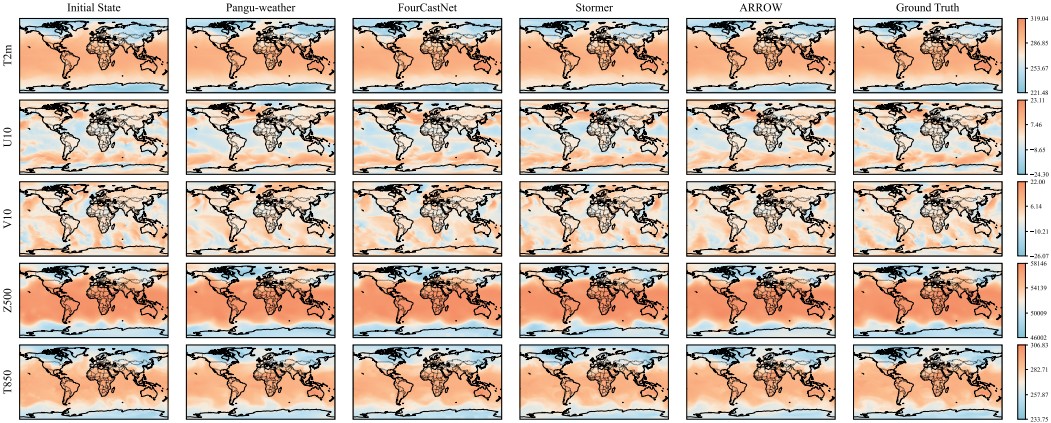

Figure 13: 222h lead time forecast comparison of different models.

We also provide forecast comparisons at other lead times to highlight ARROW's strong performance in global weather forecasting.

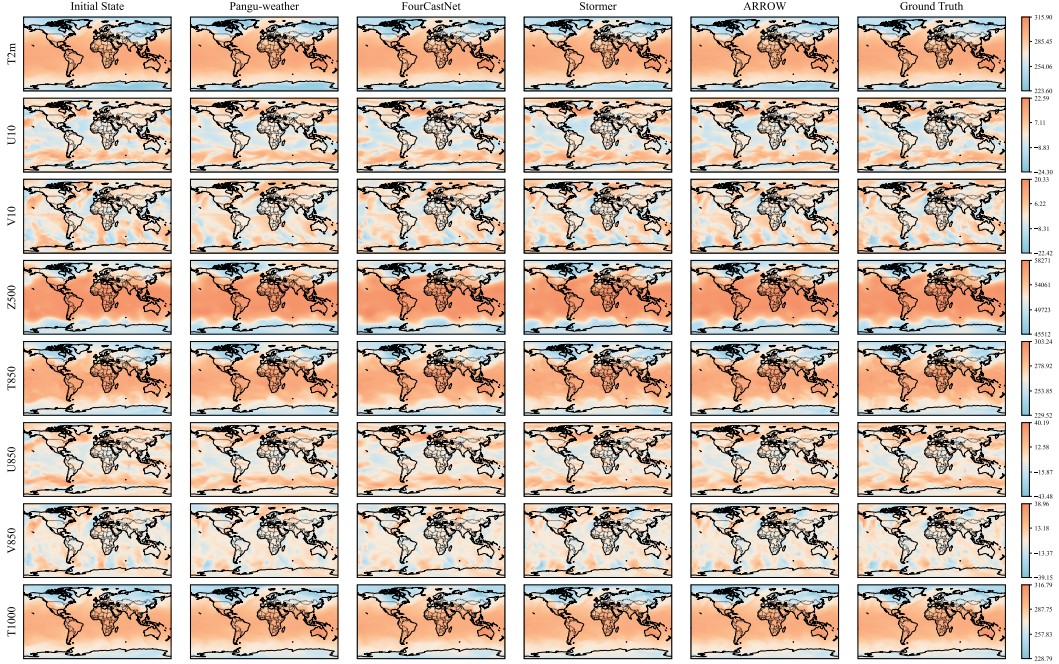

Figure 14: 126h lead time forecast comparison of different models.

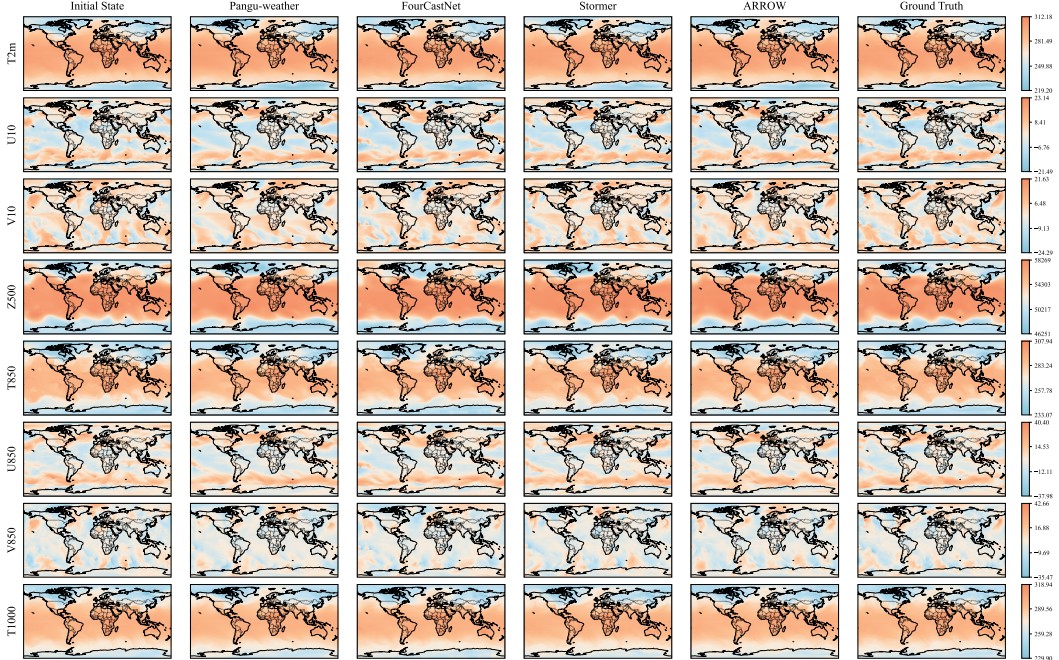

Figure 15: 222h lead time forecast comparison of different models.

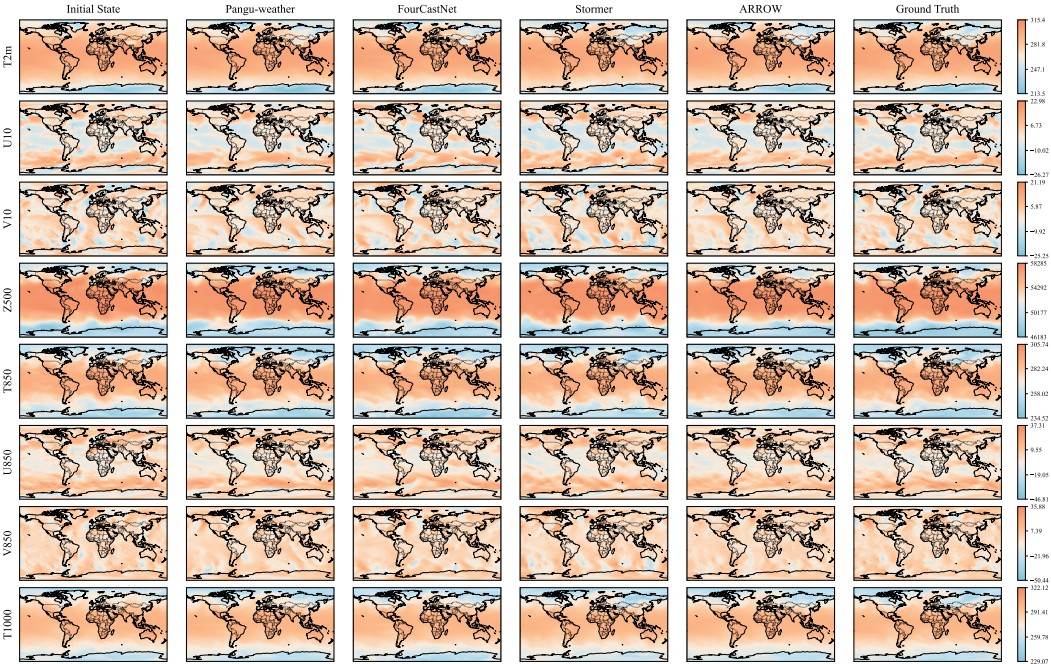

Figure 16: 354h lead time forecast comparison of different models.

# K  THE IMPORTANCE OF WEATHER PREDICTION FOR DOWNSTREAM APPLICATIONS

Time series forecasting (Yi et al., 2024; Guo et al., 2024; Jiang et al., 2025; Liu et al., 2025a; Miao et al., 2026; Zhang et al., 2025; Li et al., 2025; Wu et al., 2025; 2026) is ubiquitous in real-world applications. It typically involves predicting target variables with the support of auxiliary covari-

Table 4: Overall prediction performance comparison of other variables. The best and second-best results are highlighted in **bold** and underline, respectively.

| Methods | | Climatology | IFS | | Keisler | | GraphCast | | Pangu-weather | | FourcastNet | | Stormer | | ARROW | |
|---|---|---|---|---|---|---|---|---|---|---|---|---|---|---|---|---|
| Variable | Lead Time | RMSE↓ | RMSE↓ | ACC↑ | RMSE↓ | ACC↑ | RMSE↓ | ACC↑ | RMSE↓ | ACC↑ | RMSE↓ | ACC↑ | RMSE↓ | ACC↑ | RMSE↓ | ACC↑ |
| T500 | 5-day | | **1.85** | **0.83** | 3.00 | 0.55 | 2.16 | 0.75 | 2.39 | 0.70 | 2.81 | 0.59 | 2.00 | 0.80 | **1.85** | **0.83** |
| | 7-day | 3.163 | 2.64 | 0.64 | 3.69 | 0.34 | 2.83 | 0.56 | 3.08 | 0.50 | 3.44 | 0.39 | 2.75 | 0.61 | **2.56** | **0.65** |
| | 9-day | | 3.27 | 0.45 | 4.15 | 0.20 | 3.34 | 0.38 | 3.75 | 0.27 | 3.99 | 0.20 | 3.28 | 0.43 | **3.06** | **0.48** |
| | 14-day | | - | - | 5.06 | 0.03 | 3.96 | 0.13 | 4.19 | 0.11 | 4.29 | 0.08 | 4.41 | 0.14 | **3.69** | **0.20** |
| Z850 | 5-day | | **264.95** | **0.88** | 465.28 | 0.61 | 335.58 | 0.81 | 371.94 | 0.75 | 443.35 | 0.65 | 290.00 | 0.86 | 272.62 | 0.87 |
| | 7-day | 551.363 | 413.64 | **0.71** | 581.81 | 0.40 | 464.20 | 0.61 | 499.89 | 0.55 | 562.86 | 0.44 | 433.48 | 0.68 | **405.53** | **0.71** |
| | 9-day | | 539.45 | 0.51 | 656.10 | 0.26 | 561.58 | 0.42 | 622.02 | 0.31 | 666.70 | 0.23 | 539.66 | 0.49 | **506.17** | **0.53** |
| | 14-day | | - | - | 774.46 | 0.06 | 676.50 | 0.14 | 704.10 | 0.13 | 727.57 | 0.09 | 654.68 | 0.16 | **623.99** | **0.22** |
| U500 | 5-day | | 5.49 | **0.78** | 8.39 | 0.49 | 5.97 | 0.72 | 6.56 | 0.67 | 7.51 | 0.57 | 5.81 | 0.76 | **5.45** | **0.78** |
| | 7-day | 8.53 | 7.38 | 0.61 | 9.87 | 0.31 | 7.56 | 0.53 | 8.13 | 0.49 | 8.94 | 0.39 | 7.57 | 0.58 | **7.04** | **0.63** |
| | 9-day | | 8.88 | 0.43 | 10.97 | 0.19 | 8.71 | 0.37 | 9.66 | 0.28 | 10.23 | 0.21 | 8.84 | 0.42 | **8.24** | **0.46** |
| | 14-day | | - | - | 13.14 | 0.04 | 10.08 | 0.13 | 10.67 | 0.13 | 10.97 | 0.10 | 10.53 | 0.14 | **9.55** | **0.20** |
| V500 | 5-day | | 5.74 | **0.75** | 8.72 | 0.43 | 6.21 | 0.68 | 6.81 | 0.63 | 7.87 | 0.50 | 6.07 | 0.72 | **5.68** | **0.75** |
| | 7-day | 8.527 | 7.76 | 0.55 | 10.30 | 0.23 | 7.87 | 0.47 | 8.47 | 0.42 | 9.29 | 0.30 | 7.94 | 0.52 | **7.40** | **0.56** |
| | 9-day | | 9.28 | 0.35 | 11.30 | 0.12 | 8.99 | 0.30 | 9.97 | 0.19 | 10.40 | 0.13 | 9.22 | 0.34 | **8.57** | **0.38** |
| | 14-day | | - | - | 13.14 | 0.01 | 10.12 | 0.08 | 10.80 | 0.06 | 11.00 | 0.04 | 10.75 | 0.07 | **9.70** | **0.11** |
| U1000 | 5-day | | 3.19 | **0.81** | 5.01 | 0.37 | 3.35 | 0.66 | 3.84 | 0.58 | 4.36 | 0.48 | 3.33 | 0.71 | **3.14** | 0.74 |
| | 7-day | 4.440 | 4.20 | **0.66** | 5.83 | 0.21 | 4.20 | 0.45 | 4.64 | 0.39 | 5.04 | 0.30 | 4.27 | 0.51 | **4.00** | 0.55 |
| | 9-day | | 4.95 | **0.54** | 6.44 | 0.12 | 4.77 | 0.29 | 5.30 | 0.21 | 5.58 | 0.15 | 4.89 | 0.33 | **4.56** | 0.38 |
| | 14-day | | - | - | 7.75 | 0.03 | 5.35 | 0.09 | 5.71 | 0.09 | 5.89 | 0.07 | 5.57 | 0.10 | **5.08** | **0.16** |
| V1000 | 5-day | | 3.32 | **0.76** | 5.25 | 0.36 | 3.52 | 0.65 | 3.99 | 0.57 | 4.56 | 0.46 | 3.48 | 0.70 | **3.26** | 0.73 |
| | 7-day | 4.554 | 4.42 | **0.56** | 6.16 | 0.18 | 4.45 | 0.43 | 4.85 | 0.37 | 5.32 | 0.27 | 4.49 | 0.49 | **4.19** | 0.53 |
| | 9-day | | 5.22 | **0.39** | 6.82 | 0.09 | 5.07 | 0.26 | 5.60 | 0.16 | 5.88 | 0.11 | 5.16 | 0.30 | **4.79** | 0.34 |
| | 14-day | | - | - | 8.22 | 0.01 | 5.66 | 0.07 | 6.00 | 0.06 | 6.15 | 0.04 | 5.80 | 0.07 | **5.34** | **0.11** |

*Note:* Climatology is derived from historical averages and therefore yields a constant RMSE across all target lead times throughout the year. By definition, ACC is not applicable to Climatology. The IFS results exclude forecasts beyond 10 days as well as the TCC variable. Missing results in both cases are denoted by "–".

ates (Li et al., 2026), which often play a crucial role in forecasting accuracy. For example, wind power generation forecasting relies on local wind direction and speed, air pollutant prediction depends on temperature and humidity (Liang et al., 2023; Hettige et al., 2024; Feng et al., 2025; Tian et al., 2025), and landslide forecasting is closely associated with extreme weather conditions (Qiu et al., 2025). Consequently, meteorological forecasting acts as an upstream task for these applications, and its accuracy is critical to the performance of a wide range of downstream forecasting tasks Wang et al. (2020).

