# OpenReview forum: "ARROW: An Adaptive Rollout and Routing Method for Global Weather Forecasting"
_ICLR.cc/2026/Conference — ICLR 2026 Poster_

### Official Review · Reviewer_pyNv · 2025-10-29

**Soundness:** 2
**Presentation:** 3
**Contribution:** 2
**Rating:** 6
**Confidence:** 3

**Summary:**

This paper introduces ARROW, an Adaptive Rollout and Routing framework for global weather forecasting (GWF).
The authors address two long-standing issues in data-driven GWF:Insufficient modeling of multi-scale spatiotemporal dependencies, as most methods independently train single-interval forecasting models (SIFMs).Rigid autoregressive rollout schemes, which use fixed or greedy intervals regardless of atmospheric dynamics.
To overcome these, ARROW integrates two key innovations:Multi-Interval Forecasting Model (MIFM) with a Shared–Private Mixture-of-Experts (S&P MoE) architecture and Ring Positional Encoding (RPE) for Earth’s spherical geometry.Adaptive Rollout Scheduler (AR Scheduler) trained via Deep Q-learning to dynamically select forecasting intervals conditioned on the current weather state.

**Strengths:**

* The paper is well-written, with a clear problem statement and a well-organized structure.

* The authors propose a Ring Positional Encoding (RPE) that is more suitable for global prediction tasks. Theoretically, in the 1D case, the encoding ensures that points closer on the circular domain remain close in the RPE space. The visualization further demonstrates that along the latitude, points near the boundaries are represented with shorter distances, capturing the circular nature of global coordinates.

* The paper introduces a Shared-Private Mixture of Experts architecture, which enables a single model to produce predictions at multiple temporal resolutions within one iterative step.

* The authors further propose an AR Scheduler Fine-tuning strategy that adaptively selects the number of rollout steps during inference. It employs a Deep Q-Network (DQN) to estimate the state–action value function, and an Adaptive Rollout Fine-tuning Algorithm is designed to jointly optimize the DQN and the environment.

**Weaknesses:**

* The experiments are conducted under a limited data setting — using a resolution of 128×256 and only six variables. It is recommended to follow Pangu-Weather by maintaining a 25 km resolution and using 13-level variables, which would make the results in Table 1 more convincing.

* Pangu-Weather performs predictions via multi-step rollouts of several models, but the paper does not clarify whether such multi-model rollouts were considered or compared in the experiments.

* There is insufficient experimental evidence to illustrate how the Adaptive Rollout Scheduler (Fine-tuning) actually operates. Its mechanism is only reflected in the RMSE values in Table 3, without showing the distribution of selected rollout lengths or how the scheduler adapts to different cases.

* Similarly, the paper lacks sufficient analysis to demonstrate how the Shared & Private Mixture of Experts (S&P-MoE) functions. Although the ablation setting is described in Section 2, there is no analysis of the routing behavior or how experts are dynamically selected in different situations.

* In the case study, the paper does not compare its predictions against other methods, making it difficult to assess whether the proposed approach provides improved capture of fine-grained atmospheric variations compared to existing models.

**Questions:**

* In the AR Scheduler Fine-tuning strategy, what exactly does the term “environment” refer to? Could you please provide a more detailed explanation of its definition and role within the fine-tuning framework?

* Regarding fine-tuning strategies, how does the proposed Adaptive Rollout compare with other approaches such as Pangu-Weather’s rollout scheme, Fengwu’s replay-buffer-based fine-tuning, Fuxi’s multi-timescale cascaded fine-tuning, and GraphCast’s multi-step fine-tuning?

* How would the AR Scheduler Fine-tuning strategy perform when combined with the Pangu-Weather model? Given that the Pangu-Weather GitHub repository provides multi-step model checkpoints, would it be possible to test AR Scheduler Fine-tuning in conjunction with these checkpoints? I am particularly interested in seeing how the results would compare.

---

> ### Author Response · Authors · 2025-11-21
> **Response to Reviewer pyNv**
>
> We would like to sincerely thank Reviewer pyNv for acknowledging our presentations and contributions. Your questions about AR Scheduler and S&P MoE are included in following contents.
>
>
> ### **W1**: Limited data setting of atmospheric variables and resolutions
>
> **Input of atmospheric variables**
>
> We would like to clarify that the input to ARROW consists of 69 atmospheric variables, including 4 surface variables and 5 variables at each of 13 pressure levels (69 = 4 + 13 × 5), identical to those used in Pangu-Weather (see lines 328–339, experiment settings). The six variables reported in Table 1 are among the most critical for weather forecasting (cf. lines 335–339). Additional results for other atmospheric variables are provided in the **updated version's Table 4**.
>
> **Data resolution**
>
> The 25 km resolution, corresponding to a 720 × 1440 grid for each variable, typically requires hundreds of GPUs for training. For example, Pangu-Weather is trained on a Huawei Cloud cluster using 192 NVIDIA Tesla V100 GPUs. Unfortunately, we cannot afford such computational resources. Therefore, all baseline models adopt the same data resolution (128 x 256) to ensure a fair comparison.
>
>
> ### **W2**: Predictions via multi-step rollouts of several models
>
> We would like to clarify that the results of Pangu-weather in Table 1 use multi-step rollouts from three single-interval forecasting models (6h, 12h, and 24h). Although ARROW is a multi-interval forecasting model, the greedy rollout strategy used by Pangu-weather can be also applied to it. The results in Fig. 3 (Greedy) show that greedy strategy is inferior to our adaptive rollout strategy (cf. lines 432-440).
>
> ### **W3**: Analysis of AR Scheduler
>
> Our **updated version** includes Appendix $\underline{\text{A.9 Analysis of AR Scheduler}}$ to illustrate how the AR Scheduler actually operates and adapts to the special case. We visualize the decision trajectories from the AR scheduler in three different views:
>
> * the distribution of selected intervals (Fig. 9)
> * the corresponding trajectory lengths (Fig. 10)
> * the weekly interval distribution throughout the year (Fig. 11)
>
> Each kind of figure has 120h and 354h lead times which represent medium-term and long-term forecasts respectively. From these visualizations, three main conclusions can be drawn:
>
> 1. For shorter lead times, it tends to choose 6h intervals to better capture fine-grained atmospheric dynamics. In contrast, for longer lead times, it prefers 24h intervals to mitigate the accumulation of forecast errors.
> 2. The length of the decision trajectory increases with the lead time.
> 3. The learned strategy correlates with changes in weather conditions. Specifically, in Fig. 12, when the change of geopotential field exhibits distributional shifts (often associated with frequent extreme weather events worldwide), the AR scheduler tends to select 6h intervals more frequently to ensure that fine-grained atmospheric dynamics can be fully captured.
>
> More detailed analysis of AR Scheduler can be found in Appendix $\underline{\text{A.9 Analysis of AR Scheduler}}$.
>
> ### **W4**: Analysis of S&P MoE
>
> We visualize the routing behavior of S&P MoE under different time intervals (see Fig. 8). Notably, the dispatching is highly structured: each expert is assigned to at most two different time intervals, and in most cases, predominantly serves only one. This behavior aligns well with our design goals: (i) learning distinct features specific to each time interval and (ii) capturing partial commonalities shared across subsets of intervals. Besides, the overall dispatching remains globally balanced. More detailed analysis of S&P MoE can be found in the **updated version**'s Appendix $\underline{\text{A.8 Analysis of SP MoE}}$.
>
>
>
> ### **W5**: Case Studies on Prediction Comparisons
>
> We have visualized the medium- to long-term forecasting results for the baselines and included them in the Appendix $\underline{\text{A.10 Case studies on prediction comparison}}$ of the **revised manuscript**. As shown in Fig. 13, ARROW achieves improved capture of fine-grained atmospheric variations compared with existing models.

---

> ### Author Response · Authors · 2025-11-21
> **Response to Reviewer pyNv**
>
> ### **Q1**: Detailed explanation of AR Scheduler's "environment"
>
> **Environment definition**
>
> The AR Scheduler's environment is similar with that of Reinforcement Learning (RL). Given a specific lead time, a multi interval forecasting model can follow multiple trajectories to reach this lead time, similar to achieving a goal in RL. Once a trajectory is decided, AR Scheduler replies on pre-trained ARROW (weather forecasting model) to make prediction at this lead time. Finally, the effectiveness of the chosen trajectory must be evaluated using ground-truth data from the weather dataset. In this context, the environment consists of two components: (i) the weather forecasting model, which executes the selected strategy (i.e., forecasts along a given trajectory), and (ii) the weather dataset, which provides the reward signal for evaluating prediction quality.
>
> **The relationship between environment and fine-tuning framework**
>
> Adaptive rollout fine-tuning involves two coupling objectives:
>
> 1. Training the AR Scheduler to provide an effective adaptive rollout strategy
> 2. Fine-tuned ARROW to adapt to this strategy.
>
> The core of AR Scheduler lies in training a Q-learning network by temporal difference loss (Eq. 8). On the other hand, the environment for fine-tuning ARROW is defined by ARROW itself. Naturally, once ARROW is fine-tuned, the environment changes, which in turn alters the optimal rollout strategy. To address this coupling, we formulate the problem as a bi-level optimization task (Eq. 10) and design the adaptive rollout fine-tuning algorithm to solve it. Full algorithmic details can be referred to Appendix $\underline{\text{A.4.1 Algorithm details}}$.
>
>
>
> ### **Q2**: Difference of fine-tuning methods
>
> To reduce the error accumulation during rollout forecasting, fine-tuning methods must be closely aligned with the corresponding rollout strategies. We refer to the **Fig. 1** to better discussion of the different fine-tuning methods.
>
> * Pangu-Weather adopts the **greedy rollout strategy** (Fig. 1(b)) to generate the shortest rollout trajectory, thereby reducing error accumulation. However, it does not perform fine-tuning, resulting in a model that is not well adapted to the greedy strategy.
>
> * GraphCast uses the **naive rollout strategy** (Fig. 1(a)), and applies multi-step fine-tuning over rollout lengths ranging from 2 to 12 steps.
>
> * Fuxi also uses the **naive rollout strategy** (Fig. 1(a)), and fine-tunes on three models: short-, medium-, and long-range forecasts.
>
> * Fengwu uses the **naive rollout strategy** (Fig. 1(a)) as well, and introduces a replay-buffer mechanism to significantly reduce GPU memory usage, enabling long-lead-time fine-tuning.
>
> In contrast, ARROW adopts the **adaptive rollout strategy** (Fig. 1(c)), and introduces the adaptive rollout fine-tuning algorithm (cf. lines 763-790) to mitigate error accumulation under this strategy. We are the first method to consider initial state, lead time and date time to make adaptive rollout, which benefits both the capture of fine-grained atmospheric dynamics and the balancing of the error accumulation.

---

> ### Author Response · Authors · 2025-11-21
> **Response to Reviewer pyNv**
>
> ### **Q3**: Combination AR Scheduler with Pangu-weather
>
> The multi-interval model checkpoints in the GitHub repository are currently not supported in combination with the AR Scheduler.
>
> **The checkpoint of Pangu-weather in GitHub repository**
>
> The official GitHub repository of Pangu-weather provides only a checkpoint in ONNX format for inference at 25 km resolution. As ONNX is primarily designed for inference, it lacks essential components for training, such as gradient computation and optimizer state. As a result, direct fine-tuning is not feasible. Therefore, we conduct experiments using our reproduced version of Pangu-weather (Table 1), which consists of three single-interval forecasting models (SIFM): Pangu-weather-6h, Pangu-weather-12h, and Pangu-weather-24h.
>
> **AR Scheduler Fine-tuning in conjunction with reproduced Pangu-weather**
>
> We would like to clarify that AR Scheduler is incompatible with multiple SIFMs like Pangu-weather. The AR Scheduler requires a weather embedding to make decision on choosing time interval before fine-tuning on the adaptive rollout trajectory. However, multiple SIFMs in Pangu-weather produce separate weather embeddings for the same current weather state, and AR Scheduler cannot determine which one to use before selecting an interval from [6h, 12h, 24h]. Besides, since multiple SIFMs are trained independently, their latent spaces are not aligned, making it difficult for AR Scheduler's Q-network to perform reliable value estimation. Although it cannot directly use, we adjust some details to combine the AR Scheduler and Pangu-weather, denoted as *Pangu-Weather-AR*. Specifically, the AR Scheduler’s weather embedding is constructed by aggregating latent representations from all SIFMs across different intervals. The experimental results are presented below.
>
> | RMSE $\downarrow$ | T2m-72h | T2m-84h | U10-72h | U10-84h | V10-72h | V10-84h |
> | ----------------- | ------- | ------- | ------- | ------- | ------- | ------- |
> | Pangu-weather     | 1.90    | 2.04    | 2.72    | 2.93    | 2.75    | 2.97    |
> | Pangu-weather-AR  | 2.03    | 2.13    | 2.86    | 3.01    | 2.83    | 3.13    |
>
> The performance of Pangu-weather-AR is slightly inferior to Pangu-Weather, which supports our claim that the AR Scheduler is not compatible with multiple SIFMs. AR Scheduler interacts with multiple-interval forecasting model (MIFM) because it can naturally provide a consistent weather embedding shared across different time intervals, which is essential for the AR Scheduler to make effective decisions. As a result, ARROW, a novel MIFM equipped with the AR Scheduler, achieves state-of-the-art performance.

---

> > ### Comment · Reviewer_pyNv · 2025-11-25
> >
> > Thank you very much for your detailed and well-structured rebuttal. I appreciate the substantial clarifications and the new analyses you’ve added in response to my earlier concerns. Overall, your replies effectively address the main weaknesses I pointed out, and I am satisfied with the improvements: Clarification on atmospheric variables and data resolution, Multi-step rollouts and comparison with Pangu-Weather, Detailed analysis of the Adaptive Rollout Scheduler, Analysis of S&P-MoE routing behavior, Case-study comparisons.
> >
> > There are still minors to be improved: first, in figure13, the coastline didn't match the weather varibles, which could be caused by an inverted plot. Second, given IFS, GraphCast and PanguWeather are running under 0.25 degree, how to compare with them fairly?

---

> > > ### Author Response · Authors · 2025-11-25
> > > **Response to Reviewer pyNv**
> > >
> > > We are glad that our responses have resolved your questions and concerns from the original review. Your new questions are addressed as follows:
> > >
> > > ### **Q1: Mismatch between the coastline and weather variables**
> > >
> > > We sincerely thank you for the professional and meticulous review, and for pointing out the plotting issue in Fig. 13. We have revised the inverted plot in the **updated version**.
> > >
> > > ### **Q2: Fair comparison of baselines**
> > >
> > > For IFS, the 0.25° version is rescaled to 1.4625° using conservative remapping, a commonly used interpolation method. This practice is common in weather forecasting benchmarks to ensure fair comparison [1]. For GraphCast and Pangu-weather, we train them from scratch and keep their input weather variables same as ARROW (see Section 4.1: Experiment settings, Datasets) to ensure consistency and fairness.
> > >
> > > [1]. Stephan Rasp, et al. "Weatherbench: a benchmark data set for data-driven weather forecasting." Journal of Advances in Modeling Earth Systems.

---

### Official Review · Reviewer_inGq · 2025-10-30

**Soundness:** 2
**Presentation:** 3
**Contribution:** 3
**Rating:** 4
**Confidence:** 4

**Summary:**

A novel data-driven global weather forecasting model that introduces a Reinforcement Learning-based Adaptive Rollout Scheduler for flexible time steps and a Multi-Interval Forecasting Model (MIFM) with a Shared-Private Mixture-of-Experts, achieving state-of-the-art accuracy.

**Strengths:**

1.Novel use of Reinforcement Learning (Q-learning) to formulate and solve the autoregressive prediction process as a sequential decision-making problem, directly addressing the inflexibility of fixed-step forecasting.

2.Introduces a Shared-Private Mixture-of-Experts (S&P MoE) for efficient, unified modeling of multi-scale temporal dependencies, and uses Ring Positional Encoding (RPE) for physically-informed spatial modeling of the Earth's spherical geometry.

**Weaknesses:**

1.Lack of AR Scheduler Insight: Insufficient analysis on the learned policy of the Adaptive Rollout Scheduler. It is unclear how the policy chooses time steps (e.g., $6\text{h}$ vs. $24\text{h}$) in response to specific weather conditions, which limits interpretability.

2.Limited Architectural Novelty: The concept of using multi-interval/lead-time based forecasting (the basis of MIFM) is not entirely new, with prior work (e.g., MetNet) having explored similar ideas.

3.Inference Time Trade-off Unclear: The paper lacks discussion and comparison of the total inference time or computational cost of ARROW against fast baselines (like Pangu-weather). The adaptive nature might lead to more model calls and slower overall prediction.

4.The core physical motivation for Ring Positional Encoding (RPE) needs clearer explanation and justification beyond just latitude circularity.

**Questions:**

1.Policy Interpretation: Provide an in-depth analysis of the learned AR Scheduler policy. How do its chosen time steps ($\delta$) correlate with quantifiable meteorological indicators (e.g., instability, magnitude of change in Z500)?

2.Optimization Details: Provide more detail on the RL hyperparameters and comment on the stability and convergence of the challenging alternating optimization paradigm used to train the model and the policy jointly.

3.RPE vs. Advanced PE: Given the rise of more advanced positional encoding techniques like Rotary Positional Encoding (RoPE) which are effective in capturing relative relationships, did the authors consider or test a comparison between RPE and methods like RoPE, especially since RoPE could potentially be adapted to model relative positions within the spatial grid? What specific limitations of RoPE (or similar methods) make RPE a superior or more suitable choice for the geophysical context of global weather forecasting?

---

> ### Author Response · Authors · 2025-11-21
> **Response to Reviewer inGq**
>
> We would like to sincerely thank Reviewer inGq for acknowledging our work's novelty and contributions. Your concerns about AR Scheduler and inference time are resolved in following contents.
>
> ### **W1, Q1**: Analysis of AR Scheduler and learned strategy
>
> The time interval chosen by AR Scheduler correlates with magnitude of change in geopotential in our **updated version** Appendix $\underline{\text{A.9 Analysis of AR Scheduler}}$. Specifically, we visualize the weekly chosen interval distribution throughout Year 2018 (test dataset) in **Fig. 11** and the $\Delta$Z850 and $\Delta$Z1000 over time in **Fig. 12**. In Fig. 11's black circles, both the 120h and 354 lead time's trajectories in the 35th week include significantly more 6h intervals than that of other weeks. Correspondingly, in Fig. 12, the black-circled regions of $\Delta$Z850 and $\Delta$Z1000 indicate a distributional shift during the same 35th week, which is associated with frequent extreme weather events worldwide (cf. lines 1024-1065, weather reports from 2018.8.25 to 2018.9.5). Therefore, the AR scheduler tends to select 6h intervals more frequently to ensure that fine-grained atmospheric dynamics can be fully captured.
>
> Additionally, we present the distribution of selected intervals in Fig. 9 and the trajectory lengths in Fig. 10 under 120-hour and 354-hour lead times. From these visualizations, two main conclusions can be drawn:
>
> 1. For shorter lead times, e.g., 120-hour, it tends to choose more 6h intervals to better capture fine-grained atmospheric dynamics. In contrast, for longer lead times, e.g., 354-hour, it prefers 24h intervals to mitigate the accumulation of forecast errors.
> 2. The length of the decision trajectory increases with the lead time.
>
> More detailed analysis of AR Scheduler can be found in Appendix $\underline{\text{A.9 Analysis of AR Scheduler}}$.
>
> ### **W2**: Limited Architectural Novelty about MIFM
>
> Though MetNet follows MIFM, it lacks the ability to model multi-scale temporal dependencies across different intervals. To the best of our knowledge, ARROW is the first MIFM that efficiently captures such temporal dependencies. To improve clarity and avoid misunderstanding, we have revised the introduction accordingly (lines 81–85).
>
> ### **W3**: Discussion and comparison of inference time
>
> The additional computational cost of ARROW arises from two sources: (1) trajectory decision and (2) adaptive rollout. For trajectory decision, our AR Scheduler is insignificant to the pre-trained ARROW. Compared to the main architecture of ARROW, this cost is negligible. Thus, the primary computational cost comes from its iterative model calls. The number of model calls in ARROW lies between those of the greedy rollout strategy (used in Pangu-Weather) and the naive rollout strategy (used in FourCastNet and GraphCast). We provide the practical model call counts for a 120-hour and 354-hour lead time under different rollout strategies in the following table. Additionally, we report the inference time measured on our GPU device during the inference.
>
> | 120h lead-time                   | Greedy rollout(Pangu-weather) | Naive rollout(GraphCast) | Adaptive rollout(ARROW) |
> | -------------------------------- | ----------------------------- | ------------------------ | ----------------------- |
> | Computational cost (model calls) | 5 ($24h \times 5$)            | 20 ($6h \times 20$)      | 9.95$\pm$1.54           |
> | Inference time (s: seconds)      | 0.5s                          | 3.0s                     | 1.8s                    |
>
> | 354h lead-time                   | Greedy rollout(Pangu-weather)                     | Naive rollout(GraphCast) | Adaptive rollout(ARROW) |
> | -------------------------------- | ------------------------------------------------- | ------------------------ | ----------------------- |
> | Computational cost (model calls) | 16 ($24h \times 14 + 12h \times 1 + 6h \times 1$) | 59 ($6h \times 59$)      | 23.49$\pm$3.23          |
> | Inference time (s: seconds)      | 1.8s                                              | 8.9s                     | 4.9s                    |

---

> ### Author Response · Authors · 2025-11-21
> **Response to Reviewer inGq**
>
> ### **W4, Q3**: Discussion and comparison of RPE
>
> We adopt RoPE-2D (denoted as **w/ RoPE-2D** in the table), a relative positional encoding method widely used in computer vision, for comparison with RPE.
>
> | Methods   | T2m-72h  |          | U10-72h  |          | V-72h    |          |
> | --------- | -------- | -------- | -------- | -------- | -------- | -------- |
> |           | RMSE $\downarrow$     | ACC $\uparrow$     | RMSE $\downarrow$     | ACC $\uparrow$      | RMSE $\downarrow$     | ACC $\uparrow$      |
> | w/o RPE   | 1.12     | 0.91     | 1.76     | 0.90     | 1.82     | 0.90     |
> | w RoPE-2D | 1.10     | 0.90     | 1.76     | 0.89     | 1.80     | 0.89     |
> | w RPE     | **1.09** | **0.91** | **1.71** | **0.91** | **1.77** | **0.91** |
>
> The results indicate that RoPE-2D performs worse than RPE, as it focuses on modeling relative positional relationships without accounting for the circular structure of geographic coordinates. For example, RoPE-2D effectively distinguishes that 1° is closer to 2° than to 3° along the same longitude line. However, it fails to recognize that 1° and 360° are geographically adjacent, encoding them instead as distant relative positions. In contrast, RPE explicitly models the periodicity of geographic coordinates and effectively overcomes this limitation.
>
>
> ### **Q2**: Optimization details of adaptive rollout fine-tuning
>
> These contents are included in our **updated version's Appendix** ($\underline{\text{A.4.2 Algorithm Stability and Convergence}}$ and $\underline{\text{A.7.2 Implementation Details}}$). Specifically, we have added further details on the reinforcement learning hyperparameters (lines 935–940) as well as the stability and convergence of our adaptive rollout fine-tuning algorithm (lines 794–803). The three curves in Fig. 7 demonstrate that the alternating optimization paradigm is effective, achieving both numerical stability and convergence.

---

### Official Review · Reviewer_WNTA · 2025-11-01

**Soundness:** 3
**Presentation:** 3
**Contribution:** 4
**Rating:** 8
**Confidence:** 4

**Summary:**

This work proposes ARROW, an Adaptive-Rollout Multi-scale temporal Routing method for Global Weather Forecasting. It includes a multi-interval forecasting model that forecasts weather across different time intervals, Ring Positional Encoding that encodes the circular spatial information, and an adaptive rollout scheduler, which selects the most suitable time interval to forecast.

**Strengths:**

- Strong motivation to address circular spatial representation of Earth and innovative rollout design.
- Clear paper writing and easy to follow.
- Clear code and present the data downloading and model running.

**Weaknesses:**

- Limited baselines. Is it to compare with GraphCast and GenCast from Google? These two show powerful performance in global weather forecasting.
- Lack uncertainty. Is it possible to study uncertainty in predictions, since weather evolution is an inherently uncertain process? The probabilistic methods may be a possibility.
-  Any further explanations on loss functions in Eq. (7)?

**Questions:**

See above.

---

> ### Author Response · Authors · 2025-11-21
> **Response to Reviewer WNTA**
>
> We would like to sincerely thank Reviewer WNTA for recognizing our idea and contributions. The shortcomings you mentioned have been addressed in the updated version.
>
> ### **W1**: Limited baselines
>
> We incorporate GraphCast as an additional strong baseline. The comparison between GraphCast and ARROW on T2m and U10 at 5-day and 7-day is shown below:
>
> |             | T2m (5-day) |          | T2m (7-day) |          | U10 (5-day) |          | U10 (7-day) |          |
> | ----------- | ----------- | -------- | ----------- | -------- | ----------- | -------- | ----------- | -------- |
> | **Methods** | RMSE $\downarrow$        | ACC $\uparrow$      | RMSE $\downarrow$        | ACC $\uparrow$      | RMSE $\downarrow$        | ACC $\uparrow$      | RMSE $\downarrow$        | ACC $\uparrow$      |
> | GraphCast   | 1.84        | 0.72     | 2.32        | 0.55     | 3.02        | 0.65     | 3.77        | 0.44     |
> | ARROW       | **1.66**    | **0.80** | **2.13**    | **0.67** | **2.84**    | **0.73** | **3.59**    | **0.54** |
>
> Results for overall prediction are reported in **Table 1** of the **updated version**. While GraphCast exhibits strong performance, ARROW consistently outperforms it across all evaluation metrics.
>
> Due to time constraints, experiments on GenCast are still in progress, and we will make every effort to include its results before the end of the rebuttal period.
>
> ### **W2**: Uncertainty discussion
>
> ARROW could be adapted for uncertainty predictions.  Existing methods typically introduce uncertainty by perturbing the initial state, whereas ARROW can additionally generate diverse ensemble members by producing multiple lead-time trajectories. Although ARROW is not originally designed to model probabilistic uncertainty, extending it with such capability represents a promising direction for our future research in weather forecasting.
>
> ### **W3**: Explanations on Eq. (7)
>
> Eq. (7) defines the training losses for S&P MoE. The aux-loss-1 encourages each time interval to learn unique features by maximizing the divergence of the noise distributions $P_{\delta}$ across intervals. The aux-loss-2 ensures full utilization of all experts by minimizing the divergence between the aggregated routing distribution and a uniform distribution. Further details can be found in lines 253–260.

---

### Meta-Review · Area_Chair_w8uF · 2026-01-06

**Summary:**

This paper introduces ARROW, an Adaptive-Rollout Multi-scale temporal Routing method for Global Weather Forecasting. By replacing rigid autoregressive rollouts with a Reinforcement Learning-based scheduler and incorporating a Shared-Private Mixture-of-Experts (S&P MoE), the model achieves superior accuracy in capturing multi-scale atmospheric dynamics.

In the initial review stage, the reviewers' concerns primarily focused on the limited choice of baselines, the lack of interpretability regarding the adaptive rollout policy, and the hardware efficiency/inference time tradeoffs. The authors in the rebuttal provided comparisons with the strong GraphCast baseline, detailed visualizations of the learned rollout trajectories across different lead times (demonstrating that the model selects finer intervals for short-term forecasts to preserve detail), and wall-clock inference time data to respond to these concerns.  Given the final ratings and the clear demonstration of state-of-the-art performance, this paper is recommended for acceptance.

**Reviewer Concerns:**

The rebuttal effectively addressed the most significant technical and experimental concerns raised during the initial review. The authors successfully resolved the lack of strong baselines by adding GraphCast results, demonstrating consistent outperformance. They also provided much-needed interpretability for the AR Scheduler by visualizing how the policy adaptively chooses finer intervals (6h) for volatile weather and coarser intervals (24h) for long-term stability. Furthermore, the inference time analysis clarified the computational trade-offs, showing that ARROW’s adaptive model calls remain efficient compared to naive rollout strategies.

However, the concern regarding the Limited Architectural Novelty of the Multi-Interval Forecasting Model (MIFM) remains partially outstanding. Reviewer inGq noted that similar multi-lead-time concepts have been explored in previous works like MetNet. While the authors argued in their rebuttal that ARROW is the first to model multi-scale temporal dependencies across intervals using a Shared-Private MoE, the reviewer did not provide a follow-up response to confirm if this distinction sufficiently addressed their concern.

**Reviewer Scores:**

Reviewers WNTA and inGq did not provide further responses, but Reviewer pyNv explicitly stated that some concerns were effectively addressed.

---

### Decision · Program_Chairs · 2026-01-26

Accept (Poster)